



# Interactions between Antarctic sea ice and large-scale atmospheric modes in CMIP5 models.

Serena Schroeter[1,2,3], Will Hobbs[3,2], Nathaniel L. Bindoff[1,3,2,4]

[1]Institute for Marine and Antarctic Studies, University of Tasmania, Hobart, 7004, Australia
5  [2]Australian Research Council Centre of Excellence for Climate System Science (ARCCSS), Hobart, 7004, Australia
[3]Antarctic Climate and Ecosystems Cooperative Research Centre (ACE CRC), Hobart, 7004, Australia
[4]CSIRO Oceans and Atmosphere, Hobart, 7004, Australia

*Correspondence to*: Serena Schroeter (serena.schroeter@utas.edu.au)

**Abstract**

10  The response of Antarctic sea ice to large-scale patterns of atmospheric variability varies according to sea ice sector and season. In this study, interannual atmosphere-sea ice interactions were explored using observation-based data and compared with simulated interactions by models in the Coupled Model Intercomparison Project Phase 5. Simulated relationships between atmospheric variability and sea ice variability generally reproduced the observed relationships, though more closely during the season of sea ice advance than the season of sea ice retreat. Atmospheric influence on sea ice is known to be strongest during 15  its advance, with the ocean emerging as a dominant driver of sea ice retreat; therefore, while it appears that models are able to capture the dominance of the atmosphere during advance, simulations of ocean-atmosphere-sea ice interactions during retreat require further investigation. A large proportion of model ensemble members overestimated the relative importance of the Southern Annular Mode compared with other modes on high southern latitude climate, while the influence of tropical forcing was underestimated. This result emerged particularly strongly during the season of sea ice retreat. The amplified zonal patterns 20  of the Southern Annular Mode in many models and its exaggerated influence on sea ice overwhelm the comparatively underestimated meridional influence, suggesting that simulated sea ice variability would become more zonally symmetric as a result. Across the seasons of sea ice advance and retreat, 3 of the 5 sectors did not reveal a strong relationship with a pattern of large-scale atmospheric variability in one or both seasons, indicating that sea ice in these sectors may be influenced more strongly by atmospheric variability unexplained by the major atmospheric modes, or by heat exchange in the ocean.

25  **1. Introduction**

Antarctic sea ice extent has increased by approximately 1.5% per decade since satellite observations began in 1979 (Parkinson and Cavalieri, 2012; Turner et al., 2015a). The small overall increase masks higher-magnitude regional and seasonal trends around the continent, most notably an increase of 3.9% per decade in the Ross Sea peaking during spring, and a decrease of -3.4% per decade in the Amundsen and Bellingshausen Seas peaking during autumn (Turner et al., 2015a). By contrast, models



in the Coupled Model Intercomparison Project Phase 5 (CMIP5) exhibit decreasing sea ice trends in all months (Turner et al., 2013a). The reasons for the disparity between observed and modelled trends are not yet well understood (Bindoff et al., 2013; Hobbs et al., 2016). A large proportion of the observed trends are thought to be driven by interactions between Antarctic sea ice and atmospheric processes such as wind (Liu et al., 2004; Raphael, 2007; Lefebvre and Goosse, 2008; Massom et al., 2008;

Yuan and Li, 2008; Holland and Kwok, 2012; Matear et al., 2015), and it has been suggested that deficiencies in the model representation of atmospheric circulation may account for at least part of this disparity (Hosking et al., 2013; Mahlstein et al., 2013). The response of Antarctic sea ice to atmospheric forcing incorporates complex feedbacks and interactions between the atmosphere, sea ice and ocean (Lefebvre and Goosse, 2008; Raphael and Hobbs, 2014; Matear et al., 2015), and measuring the extent to which these feedbacks and interactions are represented in global climate simulations could provide insight into

the representation of sea ice trends and variability.

The Southern Annular Mode (SAM) is the dominant mode of atmospheric variability in the Southern Hemisphere (Gong and Wang, 1999; Limpasuvan and Hartmann, 1999; Thompson and Wallace, 2000; Marshall, 2003). It is a zonally symmetric atmospheric structure with pressure anomalies of opposing signs vacillating between the polar- and mid-latitudes of the

Southern Hemisphere (SH) (Karoly, 1990; Gong and Wang, 1999; Thompson and Wallace, 2000). The positive phase of SAM is characterised by a poleward shift and intensification of westerly circumpolar winds (Thompson et al., 2000; Marshall, 2003) which has previously been thought to increase the northward expansion (and greater areal coverage) of sea ice through Ekman transport  (Hall and Visbeck, 2002; Sen Gupta and England, 2006), while simultaneously pushing warmer oceanic air masses from the north over the comparatvely cold land of the Antarctic Peninsula (Thompson and Wallace, 2000; Marshall et al.,

2006; van Lipzig et al., 2008). A trend has been observed of the SAM moving towards its high-index (positive) polarity, with negative pressure anomalies over the Antarctic continent and positive anomalies in the mid-latitudes (Thompson et al., 2000; Thompson and Solomon, 2002; Marshall, 2003; Fogt et al., 2009). This trend is associated with stratospheric ozone depletion and forcing by greenhouse gases (Gillett and Thompson, 2003; Thompson et al., 2011; Ferreira et al., 2015). However, it has been recently suggested that the response of the Southern Ocean surface to a sustained SAM trend is more complex than the

interannual Ekman response, whereby an initial sea ice expansion is followed by warming over the longer term caused by upwelling of relatively warm, mixed-layer ocean water  (Marshall et al., 2014; Ferreira et al., 2015; Armour et al., 2016).

The Amundsen, Bellingshausen, Ross and Weddell Seas fall within a zone of orography that is non-axisymmetric, and experiences the highest mean sea level pressure variability in the SH (Lachlan-Cope et al., 2001). A climatological low-

pressure centre within the circumpolar atmospheric trough south of 60°S, known as the Amundsen Sea Low (ASL), plays a significant role in driving the advance and retreat of sea ice in this region (Hosking et al., 2013; Turner et al., 2013b; Fogt and Wovrosh, 2015; Raphael et al., 2015; Turner et al., 2015b). The depth and longitudinal location of the ASL, which influence sea ice, are in turn influenced by tropical forcing (Yuan and Martinson, 2001; Ding et al., 2011; Schneider et al., 2011; Fogt and Wovrosh, 2015; Raphael et al., 2015), radiative forcing (Fogt and Wovrosh, 2015; Raphael et al., 2015) and the phase of





the SAM (Lefebvre et al., 2004; Turner et al., 2013b). The divergent sea ice trends of the Amundsen/Bellingshausen and Ross Seas are associated with the deepening of the ASL in recent decades (Turner et al., 2013b). Recent studies have suggested that trends in the ASL and associated winds affecting sea ice in these regions are within the bounds of modelled intrinsic variability (Turner et al., 2015a; Turner et al., 2015b).

The other major modes of climate variability are the two Pacific South American modes (PSA1 and PSA2), which are associated with the high-latitude atmospheric response to ENSO (Karoly, 1989; Mo, 2000; Mo and Paegle, 2001). ENSO is teleconnected to the southern polar latitudes through meridional circulation anomalies (Harangozo, 2000), and is known to impact Antarctic sea ice (Simmonds and Jacka, 1995; Kwok and Comiso, 2002; Turner, 2004; Yuan, 2004; Simpkins et al.,

2012). However, evidence suggests that ENSO is only able to strongly influence the Antarctic climate during periods where SAM is relatively weak, or an in-phase relationship exists between the PSA modes and the SAM, such as when the warm (cold) ENSO phase coincides with a negative (positive) SAM (Fogt and Bromwich, 2006; Stammerjohn et al., 2008; Fogt et al., 2010). This enables the ENSO to project onto the SAM and the two act synergistically to enhance pressure anomalies that influence Antarctic sea ice (Karoly, 1989; Fogt and Bromwich, 2006; Stammerjohn et al., 2008; Bernades Pezza et al., 2012).

The high-latitude atmospheric response to ENSO is linked to sea ice anomalies in the Amundsen, Bellingshausen, Ross and Weddell Seas (Karoly, 1989; Harangozo, 2000; Kwok and Comiso, 2002; Yuan, 2004; Stammerjohn et al., 2008; Bernades Pezza et al., 2012).

While these large-scale atmospheric modes are clearly a strong influence on the observed variability of Antarctic sea ice,

whether the representation of atmospheric modes in CMIP5 models can explain the disparity between observed and modelled sea ice trends remains uncertain. Some observational studies have concluded that the dominant modes, SAM and ENSO, cannot account for regional Antarctic sea ice trends, and that lesser-understood large-scale modes or local processes should be investigated as alternative drivers (Liu et al., 2004; Yu et al., 2011; Hobbs et al., 2016). Other recent studies have shown that sea ice around Antarctica, except in the Amundsen, Bellingshausen and Ross Seas regions, is not in fact influenced to a great

extent by large-scale atmospheric modes, but is most impacted by synoptic weather (Matear et al., 2015; Kohyama and Hartmann, 2016). It is also unlikely that a single climate process or driver can explain all regional and seasonal sea ice trends (Lefebvre and Goosse, 2008; Holland, 2014; Raphael and Hobbs, 2014). Exploring the simulated interactions between atmospheric variability and Antarctic sea ice variability can provide further clarification as to which sectors of sea ice are most strongly influenced by large-scale atmospheric modes, and whether the strength of representation of these interactions leads

to more accurate simulations of Antarctic sea ice trends.

This study explores the extent to which global climate models reproduce large-scale patterns of atmospheric variability as well as the influence of these patterns on Antarctic sea ice variability. Previous analyses of Antarctic sea ice have generally delineated sea ice sectors by oceanographic and meteorological boundaries boundaries (Zwally et al., 1983, Figures 2-3;





Parkinson and Cavalieri, 2012, Figure 2). However, Raphael and Hobbs (2014) used spatial autocorrelation to calculate boundaries for independent sectors of Antarctic sea ice variability to define sectors where the sea ice is strongly correlated with neighbouring sea ice, indicating distinct sea ice regimes. The same study also calculated the average annual cycles of sea ice in each sector, revealing regionally distinct climatologies which, when aggregated to monthly intervals, produced seasons

of sea ice advance (March – August) and retreat (October – February).  Sea ice advance and retreat have been shown to be the key periods during which sea ice interacts with the atmosphere, and are more suitable for atmosphere-sea ice analysis than the traditional atmospheric seasons used in many studies (Stammerjohn et al., 2008; Renwick et al., 2012). Indeed, recent studies of change in Antarctic sea ice seasonality have concentrated on the seasons of annual advance, retreat and duration of sea ice coverage, with the annual sea ice season calculated between the sea ice minimum of one year to the next (February to February)

(Stammerjohn et al., 2012; Massom et al., 2013). This study uses these definitions for Antarctic sea ice to explore the observed interactions between large-scale atmospheric circulation and different sectors of Antarctic sea ice during the seasons of sea ice advance and retreat, and to compare these with simulated interactions in CMIP5 climate models in the same sectors and during the same seasons. Establishing the extent to which the CMIP5 models produce simulated atmosphere-sea ice interactions that closely reflect observed interactions provides insight into whether large-scale patterns of variability are responsible for driving

regional sea ice trends around Antarctica.

## 2. Data

Monthly Goddard-merged sea ice concentration data on a 25km x 25km grid were obtained from the National Snow and Ice Data Center for the period March 1979-February 2014 (Meier, 2015). These sea-ice data were then interpolated from their native grid onto a grid of 0.5° of longitude by 0.25° of latitude, equating to approximately 25km$^2$ at 60°S. From the regridded

data, sea ice extent (SIE) was calculated from the total ice area for each degree of longitude, bounded by the coast, and the 15% sea-ice concentration isoline. Monthly mean sea level pressure (SLP) data from the ERA-Interim global atmospheric reanalysis from March 1979-February 2014 were obtained from the European Centre for Medium-Range Weather Forecasts (available at http://apps.ecmwf.int/). ERA-Interim was chosen from the range of global atmospheric reanalysis products due to the consistency of its surface air temperature and surface temperature trend patterns with sea ice trends (Bromwich et al.,

2011; Hobbs et al., 2016). ERA-Interim reanalysis assimilates observed data sequentially in 12-hour cycles, combining new data in each cycle with a forecast model estimate of the global atmosphere and surface based on calculations from data in the previous cycle (Dee et al., 2011).

Model SIE and SLP data from the Coupled Model Intercomparison Project Phase 5 (CMIP5) (Taylor et al., 2009; Taylor et

al., 2012) were obtained from the CMIP5 multi-model ensemble archive at the Program for Climate Model Diagnosis and Intercomparison (PCMDI). The full names and modelling institutions for the models used in this study are shown in Table 1. Output from both the pre-industrial control (piControl) and 20th century (historical) experiments were used. The piControl



experiment, run for at least 500 years after the 'spin-up' period in which model conditions are stabilised, applies a prescribed pre-industrial atmosphere that does not evolve over time, enabling examination of internal variability within the models (Taylor et al., 2009). The historical experiment runs from 1850 to at least 2005, and applies evolving climate forcings including aerosol emissions, changes to atmospheric composition from greenhouse gases and solar forcing.

**3. Methods**

SLP (50°S-87°S) and SIE data for both the reanalysis between January 1979 and December 2014 and the model piControl experiment (various lengths) were detrended and cross-correlated to measure the strength of the relationship between these two variables. A two-sided t-test was used to determine statistical significance, after which any data that were not significant at the 0.05 confidence level were masked out. Autocorrelation in climate data can lead to an overestimate of statistical

significant (e.g. Zwiers and von Storch, 1995); however, the data were tested for autocorrelation and at the timescales used in this analysis no autocorrelation was found. For each sector, the sector-integrated SIE timeseries for advance (March-August) and retreat (October-February) were correlated with SLP during the same periods. The seasons were calculated using the length of each month in the season to compute a weighted average. This approach was based on that used by Raphael and Hobbs (2014) but also incorporated masking of insignificant values, weighted seasonal averaging, and the use of SLP instead of

geopotential height. This produces a proxy for the observed relationship between SIE in each sector with large-scale atmospheric variability.

Model data from the piControl experiment were chosen for this analysis to isolate the unforced variability in the models. Sector boundaries for SIE were based on those of Raphael and Hobbs (2014): East Antarctica (71°E - 163°E), Ross/Amundsen (163°E

- 250°E), Amundsen/Bellingshausen (250°E - 293°E), Weddell (293°E – 346°E) and King Hakon VII (346°E-71°E). As with the observations, sector-integrated model SIE for advance and retreat was correlated with model SLP for the same seasons, producing correlation maps. These were then pattern-correlated with the map of the reanalysis to obtain a single metric for how closely the models reproduce observed interactions between SLP and SIE. An ordinary least squares regression was applied to the SIE ensemble average of the historical members of each model between January 1979 and December 2005 to

reveal the model SIE trend for each sector and season, against which the pattern correlation values were plotted for comparison.

ERA-Interim SLP and historical model data poleward of 50°S were detrended and a square root cosine weighting was applied to compensate for the convergence of meridians towards the pole. An empirical orthogonal function (EOF) analysis was then conducted on the data to produce the three leading eigenvectors and time series of atmospheric variability in the high southern

latitudes during the seasons of ice advance and ice retreat. The leading eigenvectors display the spatial patterns of the SAM and the two PSA modes. As the two PSA modes both depict aspects of tropical teleconnections to the high latitudes, these modes were added together to create a single mode that describes the influence of tropical forcing on the Antarctic climate.





The results are thus presented as from two modes: the first mode (SAM) and the combined second and third modes (PSA). The same analysis was conducted on model historical SLP ensembles between 1979-2005, and the ensembles for each model were averaged into a single eigenvector to produce an average pattern for each model. The model average EOFs were then pattern-correlated with the corresponding EOFs of the reanalysis. The resulting correlation value for each model indicated the

extent to which the simulated pattern reflected the observed pattern for each of the eigenvectors. The percentage of variance explained by the simulated pattern was compared to the percentage of variance explained by the reanalysis. A 1:1 ratio indicated good agreement between the model and the reanalysis, with a higher or lower ratio indicating an overestimation or underestimation of the importance of that eigenvector in the model.

Lastly, the long-term climatological mean was removed from NSIDC SIE data to reveal SIE anomalies by longitude. These SIE anomalies were then cross-correlated with each EOF time series to show the relationship between changes to the amplitude of each atmospheric mode and anomalies of SIE in the advance or retreat seasons when the atmospheric influence of sea ice is known to be most important. The same analysis was conducted on model historical data, where the EOF for each ensemble member was calculated separately before the ensemble members for each model were averaged to produce a model average

eigenvector and time series to be correlated with SIE. The result from the reanalysis was compared to the results from the model correlations to determine whether the simulated influence of the leading atmospheric modes on SIE reflects the reanalysis.

## 4. Results

### 4.1 Observed Atmosphere-Sea Ice Interactions

In this section, the relationship between Antarctic sea ice and atmospheric conditions during the seasons of ice advance and retreat were examined. As previously discussed, interactions during the seasons of ice advance and retreat are the key focus of this study, as it is during these periods that the link between Antarctic sea ice to atmospheric forcing is strongest (Stammerjohn et al., 2008; Renwick et al., 2012). Sea ice in individual sectors responds to different atmospheric patterns, and the response also varies between the seasons of ice advance and retreat (Figure 1). Many of these response patterns are similar to those

found by Raphael and Hobbs (2014) upon whose approach this method is based; however, the use of seasonal weighting in this analysis (which was not included in the previous study) yielded different patterns for some sectors and seasons.

During ice advance, SIE in the Ross/Amundsen sector is negatively correlated with SLP over West Antarctica (Figure 1a). The negative correlation here indicates that increasing SIE in this sector is associated with a deepening of the atmospheric pattern shown. This negative correlation pattern persists into the retreat season (Figure 1b), but shifts towards the Ross Sea

and expands to incorporate a circumpolar component. The shape and location of the correlation pattern is indicative of an ASL component, which in its mean position is centred close to 110°W, while the circumpolar, zonally symmetric component reflects a SAM-like "see-saw" of pressure anomalies between the high- and mid-latitudes (Karoly, 1990; Gong and Wang, 1999;





Thompson and Wallace, 2000; Marshall, 2003). The longitudinal position of the ASL, which shifts towards the west during the winter and towards the east in summer (Turner et al., 2013b), is strongly influenced by the polarity of SAM and is itself a strong influence on the climate of West Antarctica (Hosking et al., 2013). Raphael et al. (2015) demonstrated the link between large-scale atmospheric circulation changes, particularly their effect on geostrophic flow, and the climatic influence of the

meridional and zonal location of the ASL. The correlations in Figures 1a and 1b indicate that sea ice in the Ross/Amundsen sector responds to surface air flow changes brought about by the ASL during the period of advance, and that the SAM dominates the sector during the period of retreat.

Correlations between SIE and SLP in the Amundsen/Bellingshausen sector during advance are almost the inverse of those in

the Ross/Amundsen sector during the same season, with positive correlations centred over the Amundsen Sea and extending from the Ross Sea towards the Bellingshausen Sea (Figure 1c). This indicates that the ASL is the dominant large-scale atmospheric driver for the Amundsen/Bellingshausen sector during the period of ice growth, and is consistent with previous analysis showing the influence of the ASL on the meridional wind field in the West Antarctic region (Hosking et al., 2013). During the retreat season, the correlation pattern remains in a similar area but contracts northwards and towards the Ross Sea

(Figure 1d). This does not follow the longitudinal shift of the ASL described above, but rather reflects the spatial pattern of the PSA (Mo and Paegle, 2001). This atmospheric pattern is generally taken to reflect the relationship between ENSO and the high latitudes, and indicates the influence of tropical forcing on sea ice in the Amundsen/Bellingshausen sector during ice retreat, in agreement with Raphael and Hobbs (2014).

In the Weddell sector during ice advance, there is no significant correlation between SIE and SLP (Figure 1e), indicating that there is no distinct large-scale atmospheric influence on sea ice in this sector and season. Rather, sea ice in this region during ice advance is more likely driven by alternative factors such as synoptic-scale weather systems, intrinsic variability, or the ocean. During retreat, the positive correlation pattern in the Weddell sector is similar to the pattern in the Ross/Amundsen sector during the same season, exhibiting an ASL component but not the zonally symmetric SAM component (Figure 1f). The

inverse sign of the correlations compared with the Ross/Amundsen sector indicates that as the atmospheric circulation pattern deepens, sea ice extent in the Weddell Sea decreases. This reflects the implied circulation of the ASL and SAM in this region, where stronger southerly winds over the Ross Sea result in the northward transport and reduced melt of sea ice in this region and stronger northerlies over the north of the Antarctic Peninsula confining ice in the Weddell Sea and increasing melt (Liu et al., 2004). The apparently differing drivers affecting ice advance and retreat in the Weddell sector agrees with recent findings

by Matear et al. (2015) that changes to sea ice in the West Atlantic region are likely driven by combined wind variability from synoptic and large-scale atmospheric patterns.

The King Hakon VII sector during advance (Figure 1g) shows a pattern that is weakly reminiscent of that observed in previous studies which have linked sea ice in this sector to the SAM (Turner et al., 2015a). However, during retreat the SAM-like pattern




disappears (Figure 1h), indicating that the region becomes more sensitive to other factors such as weather and a small ENSO forcing as suggested by Matear et al. (2015). Correlations in the East Antarctica sector do not reveal the SAM-like patterns found by Raphael and Hobbs (2014) during either advance or retreat (Figures 1i and j), but rather SIE is negatively correlated to SLP over the eastern Ross and Amundsen Seas and positively correlated to the South Atlantic during advance. During

retreat, the negative correlations shift to an area between 130°E-180°E and are stronger, while the positive correlations in the South Atlantic become negative. This agrees with previous studies showing that annual SIE in roughly this same region is influenced more by cyclonic activity around the West Pacific Ocean rather than a large-scale atmospheric pattern (Matear et al., 2015; Turner et al., 2015a).

In summary, large-scale atmospheric circulation patterns do not appear to be a dominant force in all sectors and seasons. The

ASL is the dominant force in the Ross/Amundsen and Amundsen/Bellingshausen sectors during advance and the Weddell sector during retreat, while SAM-sea ice interactions occur in the King Hakon VII sector during advance and in the Ross/Amundsen sector during ice retreat. The PSA pattern occurs in the Amundsen/Bellingshausen sector and to a smaller extent in King Hakon VII during retreat.

### 4.2 Simulated Atmosphere-Sea Ice Interactions

The analysis presented for the observations in Figure 1 was repeated for each of the models' piControl simulations. The correlations for each model were compared to the observed patterns in each sector and season to determine how closely the models represent the observed atmosphere-sea ice interactions (Figure 2). A high pattern correlation value indicates that the simulated interactions closely reflected the observed interactions, while a value near zero indicates that the two were substantially different. A high negative correlation value means that the pattern was similar, but the correlation was the inverse

sign to the observations. The correlation between simulated patterns and observed patterns during advance is plotted horizontally, while the correlations during retreat are plotted vertically for each sector. (Correlation maps for individual models can be seen in Figure S1).

Simulated SIE and SLP correlation patterns most closely reflect observed patterns during the season of advance. The

percentage of variance in the observed pattern explained by each simulated pattern can be obtained by calculating the coefficient of determination, $r^2$, which is the square of the pattern correlation value. During advance, 5 of the 16 models simulate a correlation pattern in the Ross/Amundsen sector that explains at least 80% of the spatial variance in the observed pattern, while 12 of the 16 models simulate a pattern that explains over 50% of the observed pattern. Correlations are even stronger in the Amundsen/Bellingshausen sector during retreat, with 7 of the 16 models simulating a correlation pattern that

explains at least 80% of the observed pattern, and 13 of the 16 producing a pattern explaining over 50% of the observed pattern. For East Antarctica and King Hakon VII, the number of model simulations explaining at least 50% of the observed pattern is 12 and 4 respectively.





However, during the retreat season, the simulated patterns are less consistent with the observed patterns. Only in the Weddell sector do more simulations explain over 50% of the observations in retreat (5) than in advance (0). In the Ross/Amundsen sector, simulations during ice retreat continue to reflect the observations reasonably well, though not as strongly as in advance. In the remaining sectors, especially East Antarctica and the Amundsen/Bellingshausen, the simulations largely do not capture

the observed SIE and SLP correlations during retreat.

These results have shown that the models have varying levels of success in representing the atmosphere's impact on sea ice variability. It is particularly interesting that the models reproduce these atmosphere-sea ice interactions more strongly during the period of advance than during retreat, especially given the strong representations in the Ross/Amundsen and

Amundsen/Bellingshausen sectors, and that ice advance is the period during which model trends of SIE in these sectors deviate most significantly from the observed trends (Hobbs et al., 2015; Hobbs et al., 2016). Given the discrepancy between simulated and observed SIE trends, it is pertinent to consider whether the extent to which models represent observed atmospheric variability also impacts upon their representation of sea ice trends. To examine this issue, the same pattern correlation values discussed above are plotted for each model against that model's SIE trend for that sector and season, which is calculated using

the ensemble average of the model's historical simulation (Figure 3). The observed trend for each sector and season is plotted as a red dotted line. There does not appear to be a strong relationship between either higher pattern correlation values (indicating close agreement between the model correlation maps and that of the reanalysis), or the proximity of model SIE trends to observed SIE trends in each sector and season. This is most clearly noticeable in the Ross/Amundsen, the Amundsen/Bellingshausen and East Antarctica, particularly during advance (Figures 3a, 3c, and 3i). In these sectors, although

the representation of the reanalysis correlations is generally strong, a wide spread in trend values is also evident. This suggests that the ability of the models to simulate correlations between SIE and SLP that reflect observed correlation patterns does not necessarily mean that models also produce SIE trends that reflect observed SIE trends.

### 4.3 Model representation of large-scale atmospheric modes

The leading atmospheric mode produced by the EOF analysis of ERA-Interim SLP data clearly displays the spatial pattern of

the circumpolar SAM and the associated ASL, explaining 36% of the variance in SLP during advance and just over 40% during retreat (Figure 4a and 4b). The second and third eigenvectors illustrate the spatial pattern of the PSA (Mo and Ghil, 1987). These two PSA modes were added together to produce a single mode representing the influence of tropical forcing on the high southern latitudes, in order to compare observation-based and simulated tropical impacts on sea ice (Figure 4c and 4d). The combined PSA EOF accounts for just over 27% of the variance in SLP during advance and 21% during retreat. The EOF

analysis was then conducted on the historical ensembles of each model, revealing the forced climate response of the models. Individual model EOFs can be seen in Figure S2. These were then correlated with the EOFs from ERA-Interim (Figure 5). Correlation values close to 1 indicate good representation of the spatial pattern of the observation-based atmospheric mode in the models, while values near 0 indicate little resemblance between them. A second metric was created by dividing the amount



of atmospheric variance explained by the model EOF by the amount of variance explained by the observation-based pattern, creating a ratio of the percentage of variance explained. A ratio of 1:1, which would appear on the dotted curved reference line, would indicate that the amount of variance explained by the pattern in the models is the same as the amount explained in the observation-based pattern, while a higher or lower ratio, appearing above or below the dotted reference line, would indicate

whether the model is over-representing or under-representing the influence of this atmospheric pattern.

The first EOF shows loose clustering across the ensemble members, indicating general agreement within individual models in their representation of the spatial pattern of the SAM during both ice advance and ice retreat (Figures 5a and 5b). Of the 73 individual ensemble members used in the study, 68 during advance and 45 during retreat produced a reasonable spatial pattern

of the SAM as evidenced by correlation values greater than 0.7. No ensembles during either advance or retreat obtained correlation values of 0.5 or less. In terms of the percentage of atmospheric variance explained by the simulated patterns compared with that of the observation-based pattern, the patterns of 45 ensemble members during advance and 60 during retreat account for a ratio of variance higher than the 1:1 ratio that indicates agreement with the variance explained by the reanalysis. This shows that the relative influence of SAM is overestimated in a large proportion of models, particularly during

the season of ice retreat, consistent with Haumann et al. (2014). The response of sea ice to SAM is stronger during retreat than during advance, so the amplification of the simulated influence of SAM occurs most strongly when the SAM matters most to simulated SIE.

The combined second and third EOFs show a large spread of correlations across the ensemble member representations of the

spatial pattern of the PSA during both ice advance and ice retreat (Figures 5c and 5d). The spread occurs across the ensemble members generally, and also across the ensemble members of individual models, indicating that in several models the spatial representation of the PSA lacks stability from one forced scenario to another. During advance, 32 ensemble members produce a PSA pattern with a correlation greater than 0.7, while during retreat only 21 ensemble members achieve this. Meanwhile, 29 ensembles during advance and 60 during retreat produce patterns that have correlations with the reanalysis of less than 0.5.

This indicates that a substantial proportion of ensemble members – indeed, the majority of ensemble members during retreat – do not produce a reasonable representation of tropical forcing in the high southern latitudes. Furthermore, the PSA patterns for 54 of the ensemble members during advance and 51 during retreat explain a lower percentage of atmospheric variance than the reanalysis. The overarching implication here is that for most ensembles, the SAM mode dominates atmospheric variability, creating a stronger zonal pattern than is seen in the reanalysis. The variance explained by the tropical mode is comparatively

weak in these ensembles as a result, and the simulated patterns of the PSA are generally weak representations of the observation-based PSA pattern. This is perhaps unsurprising, given that even basic ENSO characteristics are known to be weakly represented in the CMIP5 models (Guilyardi et al., 2012; Bellenger et al., 2014), and therefore the high-latitude teleconnections would be expected to be likewise underestimated.



The relative influence of these atmospheric modes on SIE in the historical ensembles as compared to ERA-Interim is shown in Figure 6. Correlations of the EOFs and SIE using piControl ensembles (not shown here) were consistent with the correlations using historical ensembles for both advance and retreat, as the detrending of historical ensembles reveals interannual variability rather than the forced response of the historical members. The correlations between SAM and SIE in historical ensembles

(Figure 6a and 6b) are somewhat overestimated compared to the reanalysis, though more strongly during retreat than during advance. The multi-model mean (shown in red) shows an overall zonal pattern with weak correlations between SAM and SIE that are largely consistent across the range of longitudes. The correlations of simulated PSA and SIE in historical ensembles (Figure 6c and 6d) are also overestimated, most noticeably during retreat when the reanalysis correlations are weaker than during advance. Once again, the multi-model mean of historical ensembles is zonal and with very weak correlations. The lack

of regional heterogeneity in the multi-model correlations indicates that the overall model response largely does not discriminate towards the sectors where the observed effect of the SAM and PSA is strongest.

**Discussion and Conclusions**

The metrics used in this study showed piControl simulations had surprisingly good skill in representing the observed atmosphere-sea ice interactions in several sectors. Interestingly, the representation of these interactions more closely reflected

observations during the season of ice advance than during retreat. The advance season is when the CMIP5 models sea ice trends diverge most significantly from observed trends, particularly in the Ross/Amundsen and Amundsen/Bellingshausen sectors where the highest-magnitude change is also observed (Hobbs et al., 2015; Hobbs et al., 2016). The results from Section 4.2 provide evidence that the models largely capture the sensitivity of sea ice to atmospheric drivers during the advance season. From earlier work, it is known that during retreat the atmosphere-to-ocean heat flux accounts for only up to 50% of the required

heating between 60-70°S (Gordon, 1981). Thus, the remainder of the melting process is likely to be from heat exchange between the deep ocean and surface waters and the absorption of solar radiation through leads in the sea ice, driving peripheral melt of floe edges. It is therefore expected that the dominant role of the atmosphere in driving sea ice variability would be diminished during retreat, consistent with our results, and that analysis atmosphere-ice interactions alone are unlikely to be sufficient to explain the observed interactions between the ocean, sea ice and atmosphere during retreat. As a major driver of

sea ice retreat, the role of the ocean in the melting of sea ice during this season warrants further scrutiny in models and observations. It has been shown that sea ice trends in some sectors during advance are driven by forcing and trends during the previous retreat season (Holland, 2014). This could explain why, despite the close representation of observed atmosphere-ice interactions during advance, simulated sea ice trends are most significantly different from the observations during advance.

It has previously been established that the observed influence of SAM and ENSO on high southern latitude climate is strongest during the late southern winter and spring (Jin and Kirtman, 2010; Simpkins et al., 2012). During the period of ice retreat as defined in this study, relationships between SAM and SIE from satellite and reanalysis data were found in the Ross/Amundsen





and Weddell sectors, and relationships between PSA and SIE were found in the Amundsen/Bellingshausen and King Hakon VII sectors. However, during retreat, historical simulations overestimated the relative importance of the SAM and PSA in terms of atmospheric variability as well as the relative influence of these modes on SIE. The variance in simulated PSA patterns is shown to be weak relative to the amplified zonal SAM pattern, with the latter appearing to dominate simulated atmospheric

variability and therefore large-scale atmosphere-sea ice interactions. If the simulated zonal atmospheric influence overwhelms the meridional influence, it follows that simulated sea ice variability would become more zonally symmetric as a result.

The absence of a strong influence of large-scale atmospheric modes in several sectors, and the generally good representation of these modes in others, indicates that while large-scale atmospheric variability is a strong and important influence on sea ice in some sectors, it is unlikely to be the dominant driver of sea ice change around all of Antarctica. Other possible drivers for

some sectors include sub-synoptic scale wind forcing (such as the variability of the Ross Sea Polyna driven by katabatic surges, drainage and barrier winds over the Ross Sea (Bromwich et al., 1998)), atmospheric variance not explained by the major modes, or the ocean. That simulated representations of atmosphere-sea ice interactions during advance which more closely reflect observed interactions does not appear to have a relationship with improved representation of sea ice trends warrants closer inspection of trends in atmospheric variability and winds, which is beyond the scope of this paper.

**Acknowledgements**

The authors thank Dr Marilyn Raphael and Dr Rob Massom for helpful comments in the preparation of this manuscript. SS was supported by the University of Tasmania/CSIRO Quantitative Marine Science Programme, and CMIP5 data management was supported by the Australian Research Council Centre of Excellence for Climate System Science. Data analysis and visualisation was performed using NCL (http://dx.doi.org/10.5065/D6WD3XH5). We acknowledge the World Climate

Research Programme's Working Group on Coupled Modeling, which is responsible for CMIP, and we thank the climate modelling groups (listed in Table 1) for producing and making available their model output. For CMIP the U.S. Department of Energy's Program for Climate Model Diagnosis and Intercomparison provides coordinating support and led development of software infrastructure in partnership with the Global Organization for Earth System Science Portals. This work was supported by the Australian Government's Cooperative Research Centres Programme through the Antarctic Climate and

Ecosystems Cooperative Research Center (ACE CRC), and contributes to AAS Project 4116.





**Table 1: Summary of models from the Coupled Model Intercomparison Project Phase 5 (CMIP5) used in the study, showing the Institution/Modelling Centre and official model name**

| Modelling Centre/Group | Model Name |
|---|---|
| Commonwealth Scientific and Industrial Research Organization (CSIRO) and Bureau of Meteorology (BOM), Australia | ACCESS1.0 (Bi et al., 2013) |
| CSIRO and BOM, Australia | ACCESS1.3 (Bi et al., 2013) |
| Beijing Climate Center, China Meteorological Administration | BCC-CSM1.1 (Xiao-Ge et al., 2013) |
| Canadian Centre for Climate Modelling and Analysis | CanESM2 (Arora et al., 2011) |
| National Center for Atmospheric Research | CCSM4 (Gent et al., 2011) |
| Community Earth System Model Contributors | CESM1-CAM5 (Neale, 2010) |
| Centre National de Recherches Météorologiques / Centre Européen de Recherche et Formation Avancée en Calcul Scientifique | CNRM-CM5 (Voldoire et al., 2013) |
| LASG, Institute of Atmospheric Physics, Chinese Academy of Sciences and CESS, Tsinghua University | FGOALS-g2 (Li et al., 2013) |
| NOAA Geophysical Fluid Dynamics Laboratory | GFDL-CM3 (Griffies et al., 2011) |
| Institut Pierre-Simon Laplace | IPSL-CM5A-LR (Mignot and Bony, 2013) |
| Institut Pierre-Simon Laplace | IPSL-CM5A-MR (Mignot and Bony, 2013) |
| Atmosphere and Ocean Research Institute (The University of Tokyo), National Institute for Environmental Studies, and Japan Agency for Marine-Earth Science and Technology | MIROC5 (Watanabe et al., 2010) |
| Max-Planck-Institut für Meteorologie (Max Planck Institute for Meteorology) | MPI-ESM-LR (Jungclaus et al., 2013) |
| Max-Planck-Institut für Meteorologie (Max Planck Institute for Meteorology) | MPI-ESM-MR (Jungclaus et al., 2013) |
| Meteorological Research Institute | MRI-CGCM3 (Yukimoto et al., 2012) |
| Norwegian Climate Centre | NorESM1-M (Bentsen et al., 2012) |





**Figure 1: Cross-correlations (significant at 95%) of observed SIE with ERA-Interim SLP from 1979-2014 during advance (a,c,e,g,i) and retreat (b,d,f,h,j). Red dotted lines indicate negative correlations, where a decrease in sea level pressure is associated with an increase in sea ice extent; blue lines indicate positive correlations, where a decrease in sea level pressure is associated with a decrease in sea ice extent. Black lines show sector boundaries.**

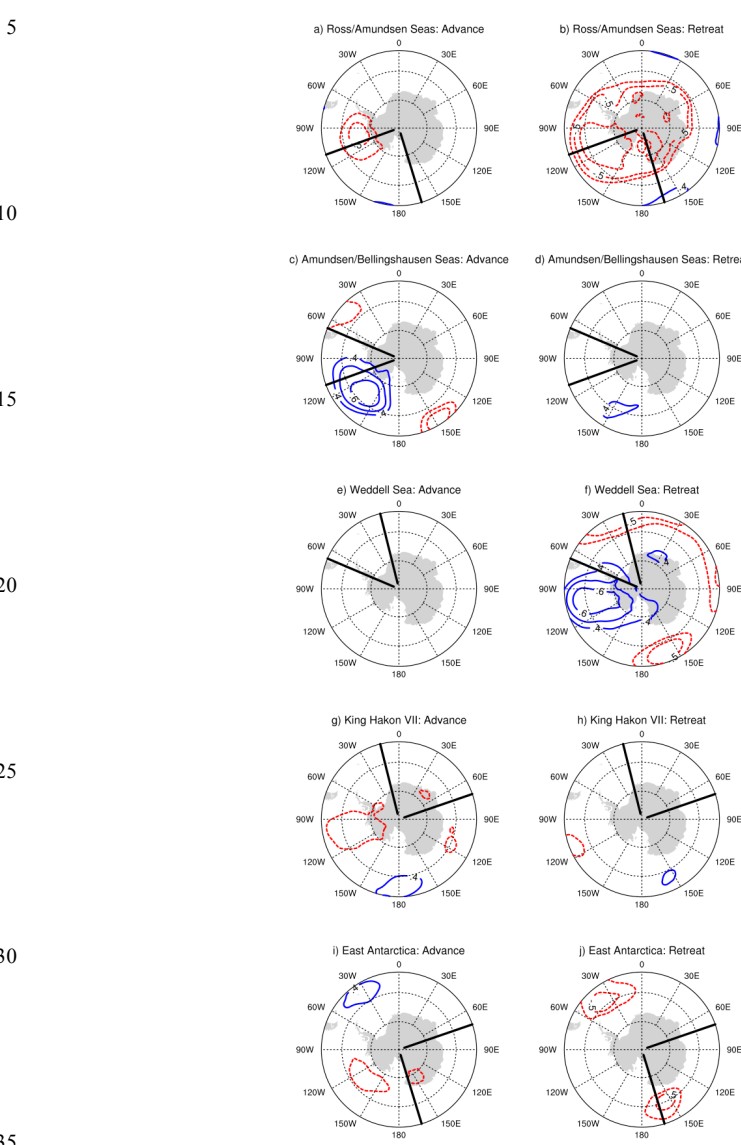





**Figure 2:** Pattern correlation values comparing observations and CMIP5 piControl ensemble correlation maps of SLP and SIE in the (a) Ross/Amundsen Seas; (b) Amundsen/Bellingshausen Seas; (c) Weddell Sea; (d) King Hakon VII; and (e) East Antarctica sectors. Dotted lines at 0.4 and -0.4 show the boundary of moderate-to-high positive and negative correlations. The diagonal line indicates where correlations for both seasons would be in agreement.

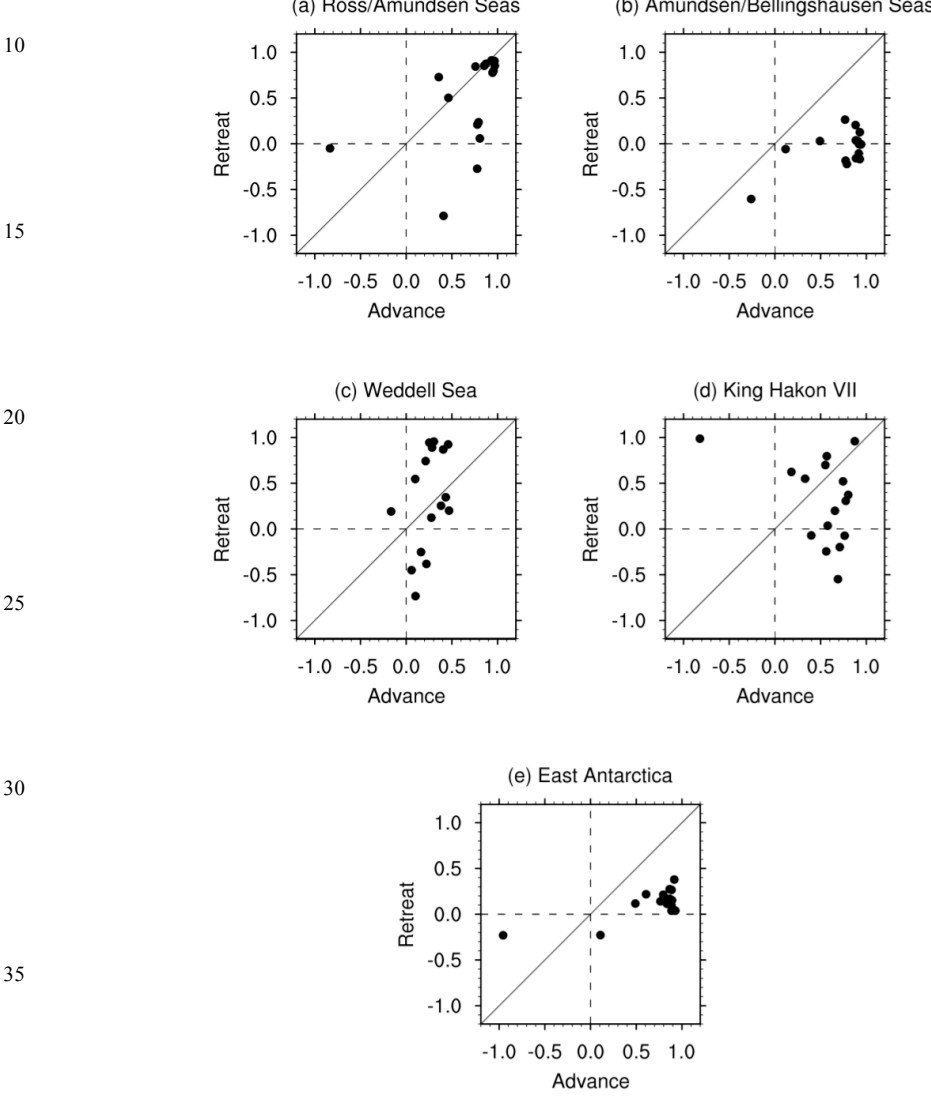





**Figure 3:** Pattern correlation values comparing observations and CMIP5 model correlation maps of SLP and SIE against the model historical (1979-2005) SIE trends for: Ross/Amundsen Seas (RAS) during (a) advance and (b) retreat, Amundsen/Bellingshausen Seas (ABS) during (c) advance and (d) retreat, Weddell Sea (WS) during (e) advance and (f) retreat, King Hakon VII (KH) during (g) advance and (h) retreat, and East Antarctica (EA) during (i) advance and (j) retreat.

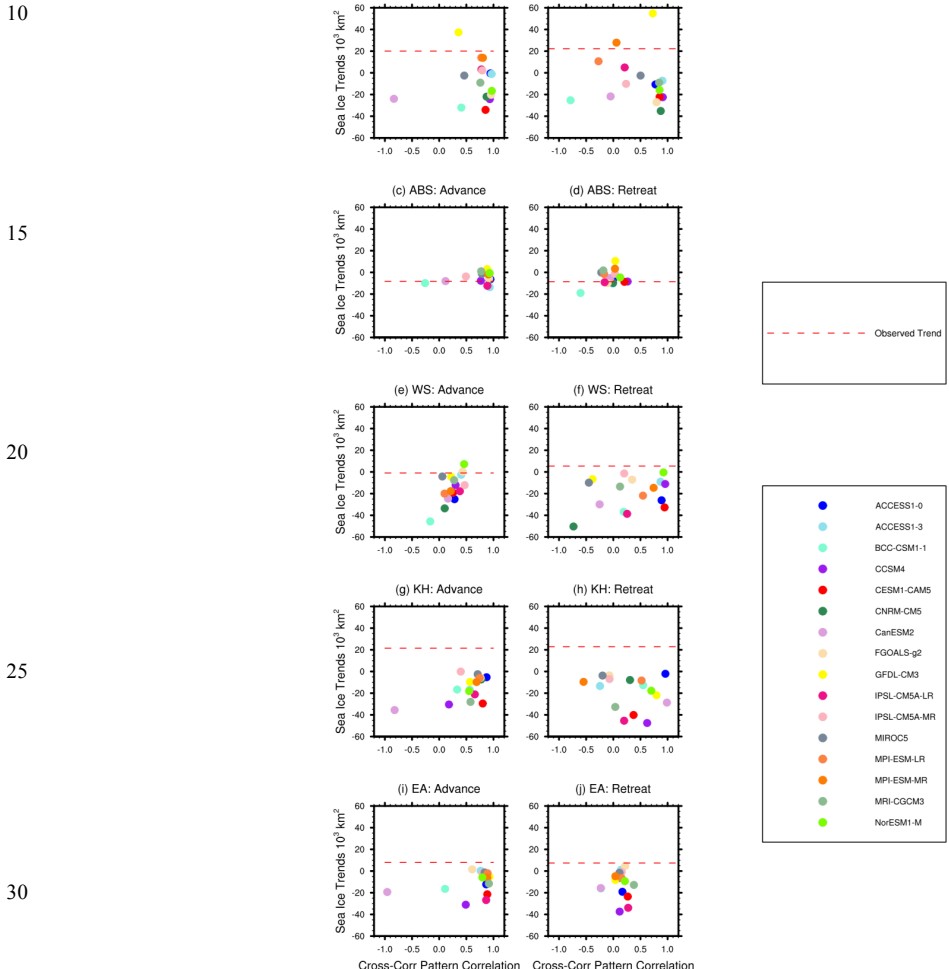

25

30



**Figure 4: Eigenvectors of ERA-Interim SLP (1979-2014) in the Southern Ocean for advance (a,c) and retreat (b,d). Numbers at top right indicate the percentage of variance in the data explained by each pattern.**

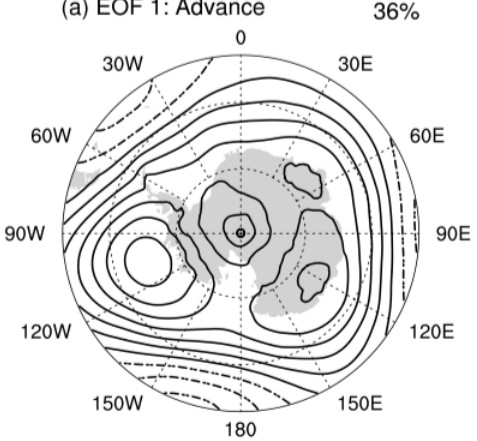
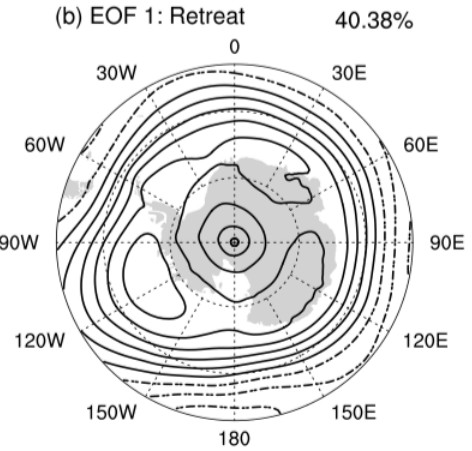

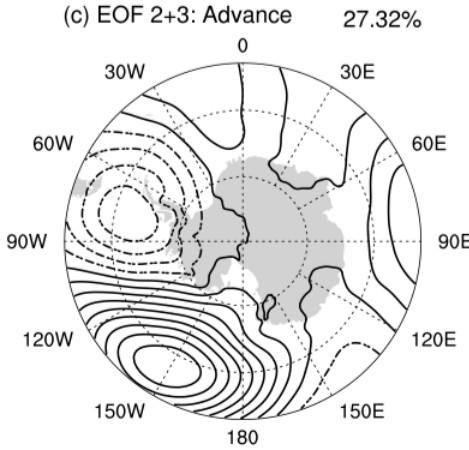
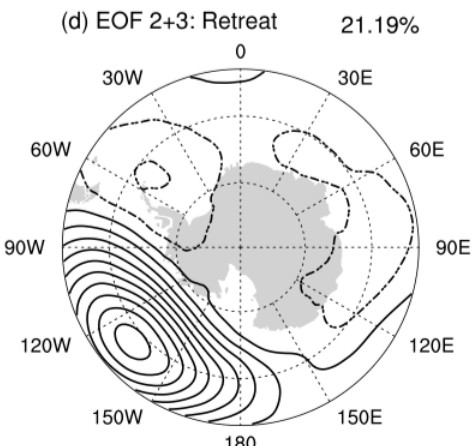





Figure 5: Taylor diagram showing the pattern correlation value (curved outer line) comparing historical CMIP5 ensemble and ERA-Interim SLP eigenvectors, and the percentage of variance explained by each pattern in the historical ensembles as a ratio of the
5   observations for EOF 1 during advance (a) and retreat (b) and the combined EOFs 2 and 3 during advance (c) and retreat (d).

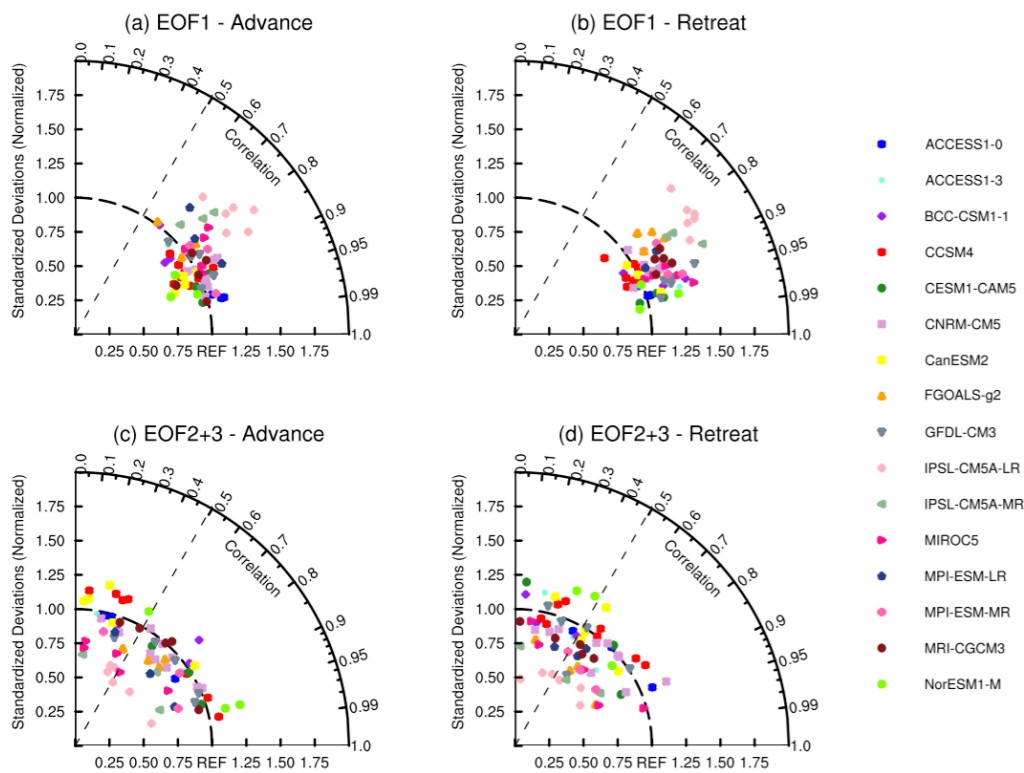





**Figure 6: Cross-correlation of SIE with historical (1979-2005) SAM during (a) advance and (b) retreat; with historical (1979-2005)**
5 **PSA during (c) advance and (d) retreat The blue line indicates observation-based correlations, with the light blue shading depicting the 95% confidence interval. The red dotted line shows the multi-model mean, and the pale grey lines show individual model correlations. The horizontal black line shows the zero line for reference.**

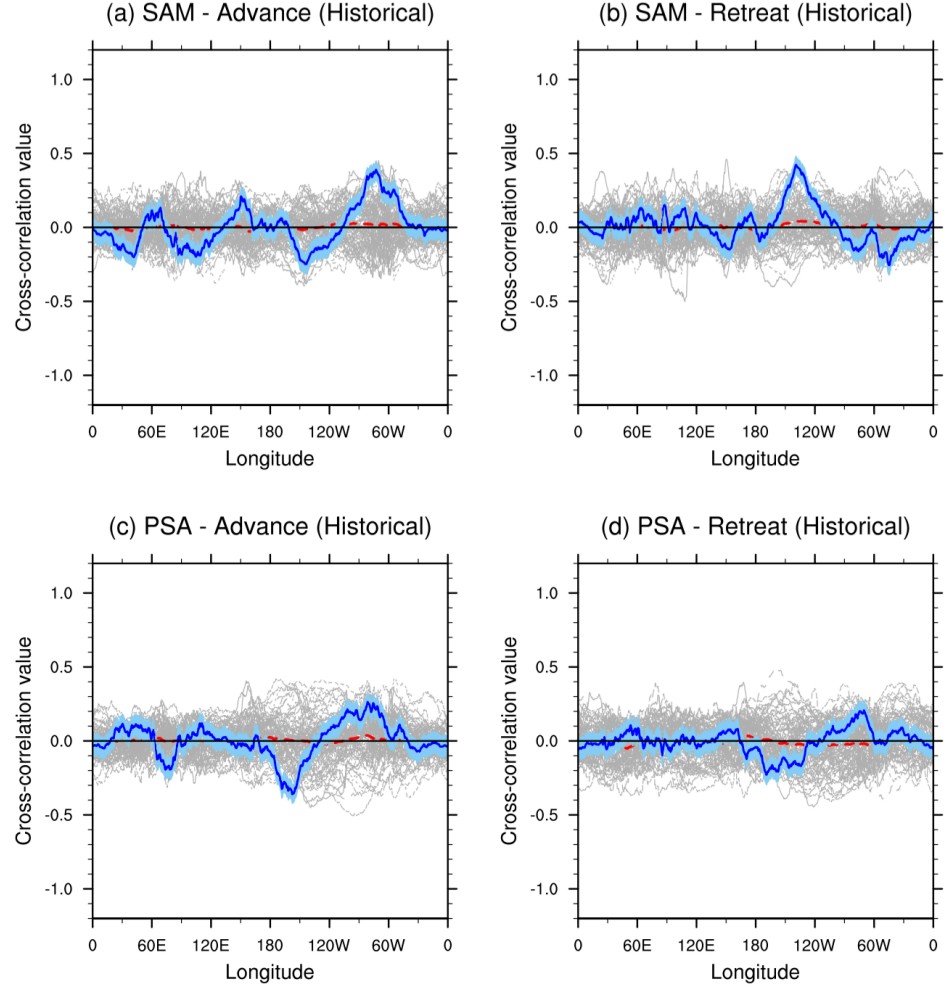





5    **Figure S1: Cross-correlations (significant at 95%) of SIE with SLP for CMIP5 historical (1979-2005) ensemble averages overlaid on observed cross-correlations during advance (a,c,e,g,i) and retreat (b,d,f,h,j). Black contours indicate observations; red dotted lines indicate negative model correlations; blue lines indicate positive model correlations. Black lines show sector boundaries.**

   **Figure S2: Individual CMIP5 model historical average eigenvectors of SLP (1979-2014) in the Southern Ocean showing the SAM**
10    **(EOF1) and the PSA (combined EOFs 2 and 3) for advance and retreat. Numbers at top right indicate the percentage of variance in the data explained by each historical average pattern.**





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
