# Peer review of "Interactions between Antarctic sea ice and large-scale atmospheric modes in CMIP5 models."

_The Cryosphere, 2016_

## Referee Comment (RC1) · Anonymous Referee #1 · 14 Nov 2016

Review of "Interactions between Antarctic sea ice and large-scale atmospheric modes in CMIP5 models" by Schroeter et al.

General Comments

This paper analyses observed Antarctic sea ice extent, sea-level pressure from the ERA-Interim reanalysis, and the CMIP5 models to assess linkages between atmospheric forcing and sea ice in both the real world and in simulations. The paper demonstrates observed linkages between sea ice variability and atmospheric forcing in different regions/seasons, and proceeds to consider how well the models capture these linkages. It also investigates how well the atmospheric variability in the models reproduces that in the reanalysis.

My overall impression of this paper is overwhelmingly positive. It contains important new material and is executed well. The results have far-reaching implications for the study of Antarctic climate, both modelled and observed. I congratulate the authors on having produced such a nice piece of work. I have a long list of suggestions for improvements in areas where I did not follow the interpretation of the results. I listed these as major revisions because there are quite a few of them, but I don't think I need to re-review the paper as I trust the authors to incorporate my points where they think appropriate.

Specific Comments

P1L15: I am unconvinced that the ocean is a dominant driver of retreat variability, and this paper shows atmospheric influences on retreat variability that are at least as important as those on advance. See comments below.

P1L19: the simulations only have an amplified SAM in terms of fraction of variability contained; the SAMs in the models could be of accurate absolute magnitude relative to observations??

P3L1: 'divergent' implied ice divergence to me

P3: There is a GRL paper in press by Kwok et al. "Linked trends in the South Pacific sea ice edge and Southern Oscillation Index" that suggests a link between SOI and the winter ice edge in the south Pacific.

P4L5: and other places: What happened to September?

P4L20: When this sentence says total ice area, it sounds like the definition of ice area (the area integral of ice concentration), not ice extent (the total area of ocean with ice concentration 15% or above). Which do the authors mean?

Section 3: I found this section very hard to follow. When I read section 4 and saw the plots, a lot of the details became clear, but only then, and I spent a lot of time trying to ingest section 3 before I moved on. For example, it was frequently unclear whether

time series were being detrended for each grid cell or for some sort of sector-wide timeseries, or whether a correlation was between a sector timeseries and a map of timeseries or another sector timeseries, etc. My suggested solution would be to only present the very basics of what data are being used in the methods section, and then to more fully explain the method underlying each figure in the results section 4.

P5L10: significance

P5L26: Why a square root cosine weighting on a grid with uniform latitude spacing?

P6L13: The EOFs from the different ensemble members are averaged together to be correlated with SIE. Which SIE? I would have thought that each ensemble member would have its own EOFs and its own SIE, so they can be directly correlated for each ensemble member?

P7L5: Is the difference in ASL-advance and SAM-retreat due to the position of the ice edge, further north at the start of retreat than it is at the start of advance?

P7L12 and others: The wording needs to be very precise. I think the finding is that the ASL is the dominant driver of *interannual variability* in sea ice advance in the A/B seas, not that it is the driver of ice advance per se. Please check this throughout the paper.

P7L21: see above! The ice in this region is definitely subjected to large-scale atmos influence, though I agree that it appears that its interannual variability is not. . .

P7L23: I do not agree that the patterns are similar.

P7L26: and SAM?

P8L11: I do not see SAM-ice interactions for Hakon.

P8L12: I do not see the PSA pattern in either sector during retreat.

Figure 2: Caption mentions lines at r=+/- 0.4 which do not appear. It would be better to

add lines showing r^2=+50% and r^2=+80%, as referred to in the text. I don't think negative values should be shown with dotted lines, since any negative correlation would be a very bad thing. Can the plot limits be set to +/-1? Can the dots be coloured like in Figure 3 so we can see which models are bad?

P8L17: It might be worth clarifying that a high correlation shows that the regional patterns are similar, but the magnitude of the relationship can still be way off in the model?

P8L25 and others, e.g page 10: I realise it is statistical convention, but the use of the word 'explained' is inappropriate here. This is just showing how well the models match the observations – the models are not explaining anything in this case.

P8L29: I think this should say 'advance' not 'retreat'

P9L16: I don't understand the 'either. . . or. . .' construction of this sentence. Is it supposed to say that there is no relationship between higher pattern correlation and veracity of model trends? Can this claim be made quantitative?

P9L22: Is the implication that the model SLP trends must be wrong? Or perhaps the model SIE and SLP patterns are spatially correlated well, but with the wrong magnitude in the correlation?

P9L30: Taking the ensemble mean EOFs does indeed reveal the forced climate response – but doesn't this complicate the comparison with ERA-Interim? The real climate is a single ensemble member, not an ensemble mean, so shouldn't ERA-interim should be compared to the population of ensemble members, not its mean?

P9L32: Similarly to the above relations between SLP and SIE, pattern correlations will reveal whether the models have a relatively strong SAM relative to the model PSA, for example, but will not detect if that SAM variability is far too weak or strong relative to the real observed SAM variability. I think this should be mentioned explicitly.

Figure 4: EOF1 explains exactly 36% of the variance in (a)?

Figure 5: I wondered if there is a concrete rationale for these being quarter-circle Taylor plots rather than just two-axis square plots like in figures 2 and 3?

P10L20: The different ensemble members' PSAs show different pattern correlations to the ERA-Interim PSA. Could this be a real result, in the sense that not just the variability but also the different modes of variability can differ between ensemble members as a result of internal variability? If so, does it make sense to judge the models too harshly against the observed PSA pattern, since that is after all just one ensemble member? If not, how does this happen in the models and not in reality?

Figure6: Could reduce the y-limits from +/- 1.2?

P11L5: I did not fully understand the argument in this paragraph. The observed relationships in Figure 6 all fit within the envelope defined by the simulations, so my default interpretation of the plot is that reality is indeed one member of the ensemble defined by CMIP5. I think the argument is that there are good physical reasons why the (single-member) observed relationships have the spatial distribution that they do (?), and this is independent of internal variability (?), so we should expect most of the simulated relationships to follow this spatial distribution (?), or perhaps at least the multi-model mean relationship should follow it (?). Also, the figure shows the envelope and mean from the simulations, but not the standard deviation, which I think is what we need to assess whether the models are wrong.

P11L8: This paragraph seemed very unclear to me and I think needs rewriting and breaking into two paragraphs. 1) The first half of the paragraph says that the models have accurate SLP-SIE relationships during advance but do not capture the observed trends during advance, but this is not explored further until a few comments at the end of the paragraph. It seems to me that this paradox could be due to either the magnitude of the SLP-SIE relation being wrong in the models (it is only a pattern correlation that is good) or the model SLP trends being wrong. The latter would be unsurprising given the poor state of the model SLP EOFs 2&3. 2) The second half of the paragraph appears to argue that in the real world the importance of atmospheric variability is diminished during retreat, but it is not (figure 1). It is the veracity of the models in reproducing atmospheric-driven ice variations that is diminished during retreat (figure 2). This could be due to model errors in any of the mechanisms mentioned, but the paragraph seems to be suggesting that the mechanisms per se reduce the effect of atmospheric variability, which is not the case. In any case, only the atmosphere-induced fraction of the variability is under consideration in this paper, not the entire variability. It may be the case that ice-climate feedbacks have an important role here. During retreat, any variability in ice cover due to winds will be amplified by melting feedbacks (e.g. albedo causes low ice to melt faster, causing lower ice). I would speculate that it is hard for models to accurately represent such feedbacks, and as a result their SLP-SIE relationships are less reliable during retreat than advance.

P12L2: I think the models underestimate the role of PSA (figure 5) in atmospheric variability?? And I am not convinced about the modelled role of PSA (figure 6).

P12L13: This sentence is worded in a very complex way and would probably be better placed in the paragraph discussed above in comment P11L8.

---

## Referee Comment (RC2) · Anonymous Referee #2 · 22 Dec 2016

General comments:

The manuscript explores the relationship between sea ice variability and large-scale atmospheric variability for the seasons of sea ice advance and sea ice retreat for five different sectors of the Southern Ocean. This is performed both for reanalysis data and model output from 16 CMIP5 models. The study provides very valuable insights into how large-scale atmospheric variability modes interact with Antarctic sea ice variability. As stated by the authors, oceanic drivers and atmospheric drivers not related to the major modes of variability are not investigated in this study. The study is clearly structured, well written and suited for this journal. I consider the manuscript almost ready for publication but want the authors to address the minor issues raised in the specific

comments.

Specific comments:

1) p.1, ll.12+:

In the abstract and the conclusion section the authors state that their paper investigates the relationship between sea ice variability and atmospheric variability. Especially in the results section however, the authors do not mention variability, but e.g. talk about "the relationship between sea ice and atmospheric conditions during the seasons of ice advance and retreat" (p.6, ll.20+). This is confusing. I am finally not sure, whether the paper really investigates the atmosphere-ice interactions in terms of variability. I encourage the authors to consistently check whether they say what they intend to say.

2) p.1, l.15:

This study does not show the ocean to be a dominant driver of sea ice retreat. The statement is hypothetical and need to be changed or removed. I like the phrasing in the final sentence of the abstract.

3) p.4, l.5:

Is there a reason why September is not considered?

4) p.4, ll.29+:

The authors mention the use of monthly reanalysis data, but they never specify the time resolution of the CMIP5 model output used. I assume this is also monthly. Please specify this here. Further the authors use reanalysis data from 1979 to 2014 but historical model output only until 2005. Why don't the authors prolong the historical simulations until 2014? At least I would like to know whether the results remain qualitatively the same when prolonging the simulations by the last 10 years, i.e. with RCP4.5.

5) p.5, ll.6+:

[Figure]

It is not clear to me how the authors detrend the reanalysis data and the piControl simulations. Did they use linear detrending for both? If so, is this appropriate for the reanalysis data? The authors should explain more specifically the methods they use.

6) p.5, ll.10-11:

Related to 4) I wonder whether monthly data is sufficient to detect autocorrelation in the SLP and SIE data.

7) The authors mention the similarity of their approach to that of Raphael and Hobbs (2014) in the method section and the similarity of theirs results to those from Raphael and Hobbs (2014) in the results section. I roughly know the study by Raphael and Hobbs (2014). However, from the present study it is not clear to me which scientific insights go beyond those from Raphael and Hobbs (2014). This needs to be pointed out more clearly. I appreciate that the authors try this distinction especially on p.4, ll.1-15, but I feel that at least its role as a predecessor study is not sufficiently accounted for.

8) p.6, ll.2-4:

I am not convinced that ensemble averaging for the historical model output is a good solution when correlating to the reanalysis. The reanalysis (and also reality) is a single realization and thus cannot be expected to be related to the ensemble average of a model.

9) p.6, section 4.1:

I have some difficulties with the description of the results presented in Fig.1.

p.7, l.14: Please mention that the correlation pattern during retreat (Fig. 1d) is much weaker than during advance (Fig.1c).

p.7, ll.23+: I do not see a pattern similarity between Fig.1b and Fig.1f, even not of inverse sign. Please check again whether the interpretation is really supported by the

results shown in Fig.1.

p.8, l.2: Why not a new paragraph for East Antarctica here?

10) p.8, l.31-32:

Are the numbers 12 for East Antarctica and 4 for King Hakon VII correct? According to Fig. 2d for King Hakon, there are more than 4 models situated above 0.5 for the advance season.

11) p.9, l.33:

The second metric is clear, but what is the first metric? This becomes not very clear by structure. Try to use the expression "the first metric" before "a second metric".

12) p.11, l.15:

The start of the sentence is misleading because to me it sounds like a definition of the advance season. I would suggest to start with: "In the advance season the modeled sea ice trends diverge ..."

13) In contrast to the rest of the manuscript, I find the conclusion section a bit weak. I think it hides some major findings that are more clearly stated in the results section. I would also love to see that the last sentence/paragraph contains the major conclusion(s) of or the overall benefit from the present paper, rather than an outlook as it is currently done. To me, this leaves the impression the results of this paper are not important which is not true.

14) Fig.1 and Fig.S1 (captions): I would prefer red dotted/blue "contours" or "isolines" instead of just "lines".

15) Fig.2 (caption): The authors mention dotted lines at 0.4 and -0.4. I cannot find them in the figure.

16) Fig.5: It would be very helpful for the reader if the authors would use the same

color for each model as in Fig.3. I cannot see a reason for not doing so.

Technical comments:

p.2, l.19: "i" is missing in comparatively

p.3, l.33: remove one "boundaries"

p.5, l.10: significance instead of "significant"

Fig.6 (caption): a dot is missing after "retreat"

---

## Author Comment (AC2) · 30 Jan 2017

In response to Reviewer 2, comment p.4 ll.29:

The authors mention the use of monthly reanalysis data, but they never specify the time resolution of the CMIP5 model output used. I assume this is also monthly. Please specify this here. Further the authors use reanalysis data from 1979 to 2014 but historical model output only until 2005. Why don't the authors prolong the historical simulations until 2014? At least I would like to know whether the results remain qualitatively the same when prolonging the simulations by the last 10 years, i.e. with RCP4.5.

The previous author response to this question was incorrectly based on the SLP-SIE

cross-correlations of the reanalysis and the simulations, which actually use piControl and not historical data. The CMIP5 historical data in question is used for EOFs, which is to what we assume this reviewer is referring. The shortened timeseries (1979-2005) has been applied to the EOFs of the reanalysis, and yielded qualitatively largely the same results as the longer timeseries (1979-2014). As such, our previous decision not to extend the historical simulations for the 73 ensembles on the basis that no substantial difference was detected between shorter and longer timeseries in the reanalysis still stands, given the additional work required to perform this. Figure S1 has been updated to show the reanalysis EOFs of the shorter timespan, not the cross-correlations.

[Figure]

**Fig. 1.** Updated Figure S1 - EOFs of ERA-Interim reanalysis SLP between 1979-2005.

---

## Author Response (AR1)

The authors would like to thank the two anonymous reviewers who have made thoughtful and insightful comments on this paper. Below, we provide a comment-by-comment response to each reviewer.

**Reviewer 1**

**P1L15: I am unconvinced that the ocean is a dominant driver of retreat variability, and this paper shows atmospheric influences on retreat variability that are at least as important as those on advance.**

Response – The sentence in the abstract has been revised to remove the reference to the ocean being a dominant driver during retreat.

From: Atmospheric influence on sea ice is known to be strongest during its advance, with the ocean emerging as a dominant driver of sea ice retreat; therefore, while it appears that models are able to capture the dominance of the atmosphere during advance, simulations of ocean-atmosphere-sea ice interactions during retreat require further investigation.

To: Atmospheric influence on sea ice is known to be strongest during its advance, and it appears that models are able to capture the dominance of the atmosphere during advance. Simulations of ocean-atmosphere-sea ice interactions during retreat, however, require further investigation.

**P1L19: the simulations only have an amplified SAM in terms of fraction of variability contained; the SAMs in the models could be of accurate absolute magnitude relative to observations??**

Response – The word 'amplified' has been removed to avoid confusion; the absolute magnitude of the models relative to the observations is now discussed in the results section.

**P3L1: 'divergent' implied ice divergence to me**

Response - 'divergent' has been replaced with 'contrasting'.

From: The divergent sea ice trends of the Amundsen/Bellingshausen and Ross Seas are associated with the deepening of the ASL in recent decades (Turner et al., 2013b).

To: The contrasting sea ice trends of the Amundsen/Bellingshausen and Ross Seas are associated with the deepening of the ASL in recent decades (Turner et al., 2013b).

**P3: There is a GRL paper in press by Kwok et al. "Linked trends in the South Pacific sea ice edge and Southern Oscillation Index" that suggests a link between SOI and the winter ice edge in the south Pacific.**

Response – reference has been included in the paper:

The high-latitude atmospheric response to ENSO is linked to sea ice anomalies in the Amundsen, Bellingshausen, Ross and Weddell Seas (Karoly, 1989; Harangozo, 2000; Kwok and Comiso, 2002; Yuan, 2004; Stammerjohn et al., 2008; Bernades Pezza et al., 2012), with recent work indicating that trends in the south Pacific ice edge during winter can be explained by changes to ice drift and surface winds resulting from a positive trend in the Southern Oscillation Index (Kwok et al., 2016).

**P4L5: and other places: What happened to September?**

Response – According to the calculations of Raphael & Hobbs (2014), sea ice in the different sectors around Antarctic stops advancing during August, instead maintaining the winter maximum throughout September before beginning its retreat in October. The only exception was the King Hakon VII sector which reached its maximum later than the others and began its retreat one month later; however, to compare like with like, we used the majority advance period for all sectors in this study. None of the sectors had an extended minimum, which is why the end of retreat and start of advance do not have a gap.

**P4L20: When this sentence says total ice area, it sounds like the definition of ice area (the area integral of ice concentration), not ice extent (the total area of ocean with ice concentration 15% or above). Which do the authors mean?**

Response – We mean sea ice extent here; as stated in the manuscript, we use the 15% sea ice concentration isoline. "Area" has been replaced in the text by "cover" to avoid confusion.

From: From the regridded data, sea ice extent (SIE) was calculated from the total ice area for each degree of longitude, bounded by the coast, and the 15% sea-ice concentration isoline.

To: From the regridded data, sea ice extent (SIE) was calculated from the total sea ice cover for each degree of longitude, bounded by the coast, and the 15% sea ice concentration isoline.

Section 3: I found this section very hard to follow. When I read section 4 and saw the plots, a lot of the details became clear, but only then, and I spent a lot of time trying to ingest section 3 before I moved on. For example, it was frequently unclear whether time series were being detrended for each grid cell or for some sort of sector-wide timeseries, or whether a correlation was between a sector timeseries and a map of timeseries or another sector timeseries, etc. My suggested solution would be to only present the very basics of what data are being used in the methods section, and then to more fully explain the method underlying each figure in the results section 4.

Response – The method section has been substantially revised to more clearly explain the steps taken for each part of the analysis in order to reduce confusion.

**P5L10: significance**

Response – This has been updated in the manuscript.

**P5L26: Why a square root cosine weighting on a grid with uniform latitude spacing?**

Response – As stated in the manuscript, cosine weighting is used to account for the change of longitude distance with latitude. The cosine weighting is essentially an areal weighting, thus each grid cell has equal influence in the EOF analysis.

**P6L13: The EOFs from the different ensemble members are averaged together to be correlated with SIE. Which SIE? I would have thought that each ensemble member would have its own EOFs and its own SIE, so they can be directly correlated for each ensemble member?**

Response – This is an error in the manuscript. The individual model plots in Figure S2 should show the individual ensemble member EOFS. The Taylor diagram does actually show the individual ensemble member EOF against the same ensemble member SIE, not the model average as written. The text has been changed to reflect this, and Figure S2 has also been updated.

**P7L5: Is the difference in ASL-advance and SAM-retreat due to the position of the ice edge, further north at the start of retreat than it is at the start of advance?**

Response – This is an interesting idea; probably only answerable by looking at extensively at patterns of sea ice concentration rather than extent. This is beyond the scope of this paper, but is worthy of further analysis.

**P7L12 and others: The wording needs to be very precise. I think the finding is that the ASL is the dominant driver of \*interannual variability\* in sea ice advance in the A/B seas, not that it is the driver of ice advance per se. Please check this throughout the paper.**

Response – The wording in the paragraph has been changed to reflect this, and it has been checked throughout the paper.

[revised manuscript text omitted]

**Figure 2: Caption mentions lines at r=+/-0.4 which do not appear. It would be better to add lines showing $r^2=+50\%$ and $r^2=+80\%$ , as referred to in the text. I don't think negative values should be shown with dotted lines, since any negative correlation would be a very bad thing. Can the plot limits be set to +/-1? Can the dots be coloured like in Figure 3 so we can see which models are bad?**

Response – Lines have been revised to show r = 0.7 and 0.8. Plot limits have been changed to +/-1.0. Dots have been coloured using the same colour scheme as in Figure 3.

**P8L17: It might be worth clarifying that a high correlation shows that the regional patterns are similar, but the magnitude of the relationship can still be way off in the model?**

Response – following sentence has been added to to this paragraph for clarification.

'These comparisons only measure the extent to which the observed spatial pattern was replicated in the models, not whether the magnitude of the interactions in the models is similar to that of the observations.'

**P8L25 and others, e.g page 10: I realise it is statistical convention, but the use of the word 'explained' is inappropriate here. This is just showing how well the models match the observations – the models are not explaining anything in this case.**

Response – In this case, the 'explained' refers to the amount of variance in the data to which each pattern corresponds. The comparison between the models and the observation is only to show the difference between how much variance in one is 'explained' by a particular pattern compared to how much is 'explained' by another. However, this has been replaced in the text by the term 'accounted for' to try to avoid confusion.

**P8L29: I think this should say 'advance' not 'retreat'**

Response - Correct; this has now been updated in the text.

P9L16: I don't understand the 'either. . . or. . .' construction of this sentence. Is it supposed to say that there is no relationship between higher pattern correlation and veracity of model trends? Can this claim be made quantitative?

Response – The text has been updated to remove the ambiguity. The intention of the sentence was to explain that having a better representation of atmosphere-ice interactions does not necessarily mean the same model will also produce a sea ice trend closer to observed trends.

*From:* There does not appear to be a strong relationship between either higher pattern correlation values (indicating close agreement between the model correlation maps and that of the reanalysis), or the proximity of model SIE trends to observed SIE trends in each sector and season.

To: There does not appear to be a strong relationship between higher pattern correlation values (indicating close agreement between the model correlation maps and that of the reanalysis) and the proximity of model SIE trends to observed SIE trends in each sector and season.

**P9L22: Is the implication that the model SLP trends must be wrong? Or perhaps the model SIE and SLP patterns are spatially correlated well, but with the wrong magnitude in the correlation?**

Response – The intention of this sentence is merely to point out that if a model produces a reasonable representation of interannual variability in the relationship between SIE and SLP, it doesn't necessarily also produce a reasonable sea ice trend. The sentence has been replaced for clarification.

From: This suggests that the ability of the models to simulate correlations between SIE and SLP that reflect observed correlation patterns does not necessarily mean that models also produce SIE trends that reflect observed SIE trends.

To: These results suggest that a model with an interannual sea ice-atmosphere interaction pattern that closely represents the observed pattern will not necessarily also produce realistic sea ice trends.

**P9L30: Taking the ensemble mean EOFs does indeed reveal the forced climate response – but doesn't this complicate the comparison with ERA-Interim? The real climate is a single ensemble member, not an ensemble mean, so shouldn't ERA-interim should be compared to the population of ensemble members, not its mean?**

Response – This paragraph has been updated to more clearly state that the individual members are indeed used, not the ensemble mean.

*From: The EOF analysis was then conducted on the historical ensembles of each model, revealing the forced climate response of the models.*

To: The EOF analysis was then conducted on the individual ensemble members of each model, revealing the forced climate response of each model member.

P9L32: Similarly to the above relations between SLP and SIE, pattern correlations will reveal whether the models have a relatively strong SAM relative to the model PSA, for example, but will not detect if that SAM variability is far too weak or strong relative to the real observed SAM variability. I think this should be mentioned explicitly.

Response – We have examined the absolute variance of the principal component corresponding to each EOF (Figure 6). This shows that the model SAM and PSA variability is far too weak compared to observed variability. This has been added to the text.

**Figure 4: EOF1 explains exactly 36% of the variance in (a)?**

Response – We have rounded the variance-explained to 2 significant figures. Since this is a gross empirical metric, we believe that this is a suitable level of precision to report.

**Figure 5: I wondered if there is a concrete rationale for these being quarter-circle Taylor plots rather than just two-axis square plots like in figures 2 and 3?**

Response – Originally, two-axis plots were used to show both these metrics; however, it was not easy to see the spread among the model ensemble members in both directions using this type of plot. The authors decided instead to use a Taylor diagram, which is a clearer method of comparing the outcome of multiple ensemble members at once using the two different metrics.

P10L20: The different ensemble members' PSAs show different pattern correlations to the ERA-Interim PSA. Could this be a real result, in the sense that not just the variability but also the different modes of variability can differ between ensemble members as a result of internal variability? If so, does it make sense to judge the models too harshly against the observed PSA pattern, since that is after all just one ensemble member? If not, how does this happen in the models and not in reality?

Response – The differences between the spatial representation of the PSA modes in the difference ensemble members suggest that they may change upon multidecadal timescales. We have updated the text here to include this caveat.

**Figure6: Could reduce the y-limits from +/- 1.2?**

Response – Limits have been reduced to +/- 1.0.

P11L5: I did not fully understand the argument in this paragraph. The observed relationships in Figure 6 all fit within the envelope defined by the simulations, so my default interpretation of the plot is that reality is indeed one member of the ensemble defined by CMIP5. I think the argument is that there are good physical reasons why the (singlemember) observed relationships have the spatial distribution that they do (?), and this is independent of internal variability (?), so we should expect most of the simulated relationships to follow this spatial distribution (?), or perhaps at least the multi-model mean relationship should follow it (?). Also, the figure shows the envelope and mean from the simulations, but not the standard deviation, which I think is what we need to assess whether the models are wrong.

Response – We have added lines depicting the 1.96 standard deviation (equivalent to the 95% confidence interval for a Normally distributed ensemble for infinite degrees of freedom.

Note that given: a) the strength of the observed correlation pattern (which is large compared to the standard deviation), b) its acknowledged importance in the literature, and c) the length of correlation period (approximately 3 decades), it is highly unlikely that the differences between ensemble members could be explained by internal variability alone.

P11L8: This paragraph seemed very unclear to me and I think needs rewriting and breaking into two paragraphs. 1) The first half of the paragraph says that the models have accurate SLP-SIE relationships during advance but do not capture the observed trends during advance, but this is not explored further until a few comments at the end of the paragraph. It seems to me that this paradox could be due to either the magnitude of the SLP-SIE relation being wrong in the models (it is only a pattern correlation that is good) or the model SLP trends being wrong. The latter would be unsurprising given the poor state of the model SLP EOFs 2&3. 2) The second half of the paragraph appears to argue that in the real world the importance of atmospheric variability is diminished during retreat, but it is not (figure 1). It is the veracity of the models in reproducing atmospheric-driven ice variations that is diminished during retreat (figure 2). This could be due to model errors in any of the mechanisms mentioned, but the paragraph seems to be suggesting that the mechanisms per se reduce the effect of atmospheric variability, which is not the case. In any case, only the atmosphere-induced fraction of the variability is under consideration in this paper, not the entire variability. It may be the case that ice-climate feedbacks have an important role here. During retreat, any variability in ice cover due to winds will be amplified by melting feedbacks (e.g. albedo causes low ice to melt faster, causing lower ice). I would speculate that it is hard for models to accurately represent such feedbacks, and as a result their SLP-SIE relationships are less reliable during retreat than advance.

Response – The paragraph has been broken into two sections as suggested, with the advance sea ice-atmosphere interactions discussed first and then retreat separately to avoid confusion. It is true that complex ice-ocean feedbacks are probably difficult for models to represent; however, those ice-ocean feedbacks are equally complex (if not more so than) during advance (for example, the entrainment of sub-mixed layer into the mixed layer from brine rejection). Although incredibly important, those feedbacks don't therefore explain why advance should necessarily be better represented than retreat in the models.

**P12L2: I think the models underestimate the role of PSA (figure 5) in atmospheric variability?? And I am not convinced about the modelled role of PSA (figure 6).**

Response – The sentence has been updated to reflect the underestimation of the PSA.

From: However, during retreat, historical simulations overestimated the relative importance of the SAM and PSA in terms of atmospheric variability as well as the relative influence of these modes on SIE.

To: However, during both sea ice advance and retreat, the majority of historical simulations overestimated the relative importance of the SAM and underestimated that of the PSA.

**P12L13: This sentence is worded in a very complex way and would probably be better placed in the paragraph discussed above in comment P11L8.**

Response – This sentence has been reworded to reduce its complexity.

**Reviewer 2 – Variability**

p.1, II.12+: In the abstract and the conclusion section the authors state that their paper investigates the relationship between sea ice variability and atmospheric variability. Especially in the results section however, the authors do not mention variability, but e.g. talk about "the relationship between sea ice and atmospheric conditions during the seasons of ice advance and retreat" (p.6, II.20+). This is confusing. I am finally not sure, whether the paper really investigates the atmosphere-ice interactions in terms of variability. I encourage the authors to consistently check whether they say what they intend to say.

Response – Through addressing the comments of Reviewer 1, the text throughout the results has been revised to more specifically discuss interannual variability, and this hopefully reduces the confusion in the rest of the paper. The revised results and discussion are quite clear that the paper is discussing variability rather than trends.

**p.1, l.15: This study does not show the ocean to be a dominant driver of sea ice retreat. The statement is hypothetical and need to be changed or removed. I like the phrasing in the final sentence of the abstract.**

Response – The sentence in the abstract has been revised to remove the reference to the ocean being a dominant driver during retreat.

From: Atmospheric influence on sea ice is known to be strongest during its advance, with the ocean emerging as a dominant driver of sea ice retreat; therefore, while it appears that models are able to capture the dominance of the atmosphere during advance, simulations of ocean-atmosphere-sea ice interactions during retreat require further investigation.

To: Atmospheric influence on sea ice is known to be strongest during its advance, and it appears that models are able to capture the dominance of the atmosphere during advance. Simulations of ocean-atmosphere-sea ice interactions during retreat, however, require further investigation.

**p.4, I.5: Is there a reason why September is not considered?**

Response – According to the calculations of Raphael & Hobbs (2014), sea ice in the different sectors around Antarctic stops advancing during August, instead maintaining the winter maximum throughout September before beginning its retreat in October. The only exception was the King Hakon VII sector which reached its maximum later than the others and began its retreat one month later; however, to compare like with like, we used the majority advance period for all sectors in this study. None of the sectors had an extended minimum, which is why the end of retreat and start of advance do not have a gap.

p.4, II.29+: The authors mention the use of monthly reanalysis data, but they never specify the time resolution of the CMIP5 model output used. I assume this is also monthly. Please specify this here. Further the authors use reanalysis data from 1979 to 2014 but historical model output only until 2005. Why don't the authors prolong the historical simulations

**until 2014? At least I would like to know whether the results remain qualitatively the same when prolonging the simulations by the last 10 years, i.e. with RCP4.5.**

Response – The method section has been updated to clearly state that CMIP5 monthly historical data is used. Cross-correlations between ERA-Interim reanalysis SLP and NSIDC sea ice extent have been run also for the period January 1979-December 2005, and have yielded largely the same results (now attached as Figure S1). There is no substantial difference between the shorter and longer timeseries in the observations. Therefore, we decided not to lengthen the historical ensembles, given the significant extra work required to concatenate RCP4.5 onto 73 historical ensembles.

**p.5, II.6+: It is not clear to me how the authors detrend the reanalysis data and the piControl simulations. Did they use linear detrending for both? If so, is this appropriate for the reanalysis data? The authors should explain more specifically the methods they use.**

Response – We use the same linear detrending method for all datasets, including the reanalysis dataset, as it has been used widely both for ERA-Interim and other datasets, for example:

- Bracegirdle, T. J.: Climatology and recent increase of westerly winds over the Amundsen Sea derived from six reanalyses, International Journal of Climatology, 33, 843-851, 2013.
- Bromwich, D. H., Nicolas, J. P., Monaghan, A. J., Lazzara, M. A., Keller, L. M., Weidner, G. A., and Wilson, A. B.: Central West Antarctica among the most rapidly warming regions on Earth, Nature Geoscience, 6, 139-145, 2013.

**p.5, II.10-11: Related to 4) I wonder whether monthly data is sufficient to detect autocorrelation in the SLP and SIE data.**

Response – We're only interested in autocorrelation in this case insofar as it affects statistical significance tests. As we are using monthly data in the study, it is only appropriate to consider the autocorrelation at monthly timescales.

The authors mention the similarity of their approach to that of Raphael and Hobbs (2014) in the method section and the similarity of theirs results to those from Raphael and Hobbs (2014) in the results section. I roughly know the study by Raphael and Hobbs (2014). However, from the present study it is not clear to me which scientific insights go beyond those from Raphael and Hobbs (2014). This needs to be pointed out more clearly. I appreciate that the authors try this distinction especially on p.4, II.1- 15, but I feel that at least its role as a predecessor study is not sufficiently accounted for.

Response – We have updated the text to clearly refer to the Raphael and Hobbs (2014) study in the Introductory section, Method section and Discussion so as to most clearly differentiate between the predecessor study and this one.

**p.6, II.2-4: I am not convinced that ensemble averaging for the historical model output is a good solution when correlating to the reanalysis. The reanalysis (and also reality) is a**

**single realization and thus cannot be expected to be related to the ensemble average of a model.**

Response – This is an error in the manuscript. The individual model plots in Figure S2 should show the individual ensemble member EOFS. The Taylor diagram does actually show the individual ensemble member EOF against the same ensemble member SIE, not the model average as written. The text has been changed to reflect this, and Figure S2 has also been updated.

**p.6, section 4.1: I have some difficulties with the description of the results presented in Fig.1.**

**p.7, l.14: Please mention that the correlation pattern during retreat (Fig. 1d) is much weaker than during advance (Fig.1c).**

Response – The text has been updated to include the weakening of the pattern.

From: During the retreat season, the correlation pattern remains in a similar area but contracts northwards and towards the Ross Sea (Figure 1d).

To: During the retreat season, the correlation pattern remains in a similar area but weakens, contracting northwards and towards the Ross Sea (Figure 1d).

**p.7, ll.23+: I do not see a pattern similarity between Fig.1b and Fig.1f, even not of inverse sign. Please check again whether the interpretation is really supported by the results shown in Fig.1.**

Response – As described in the text, the correlation pattern for Figure 1f shows the nonannular component of the SAM, the ASL, but not the annular component of the ASL. The ASL pattern is inverse to that of the non-annular component of the SAM pattern in the Ross/Amundsen sector during the same sector. Physically, this is well known, as the implied circulation of the ASL results in increased southerly winds over the Ross Sea and northerly winds over the Antarctic Peninsula and Weddell Sea (e.g. Turner et al. 2015).

**p.8, l.2: Why not a new paragraph for East Antarctica here?**

Response – A separate paragraph has been inserted here for East Antarctica.

**p.8, I.31-32: Are the numbers 12 for East Antarctica and 4 for King Hakon VII correct? According to Fig. 2d for King Hakon, there are more than 4 models situated above 0.5 for the advance season.**

Response – We have double checked this, and altered the Figure to make the results clearer as a result of comments from Reviewer 1. These numbers are indeed correct; in order for the model to obtain an  $r^2$  score of 50%, it needs to have a pattern correlation of 0.7 or higher, not 0.5 (which would only obtain an  $r^2$  score of 25%). Dotted lines at 0.7 and 0.8 have been added at the urging of Reviewer 1 in order to make this clearer, and we hope this assists with the comments of Reviewer 2 as well.

**p.9, l.33: The second metric is clear, but what is the first metric? This becomes not very clear by structure. Try to use the expression "the first metric" before "a second metric".**

Response – The text has been updated here to avoid confusion.

From: Correlation values close to 1 indicate good representation of the spatial pattern of the observation-based atmospheric mode in the models, while values near 0 indicate little resemblance between them. A second metric was created by dividing the amount of atmospheric variance explained by the model EOF by the amount of variance explained by the observation-based pattern, creating a ratio of the percentage of variance explained.

To: The results are explained using two metrics. The first metric, correlation values, is used to indicate the strength of the simulated representation of the spatial pattern seen in the reanalysis. A correlation value close to 1 indicates good representation of the pattern, while values near 0 indicate little resemblance between the two. A second metric was created by dividing the amount of atmospheric variance explained by the model EOF by the amount of variance explained by the observation-based pattern, creating a ratio of the percentage of variance explained.

**p.11, I.15: The start of the sentence is misleading because to me it sounds like a definition of the advance season. I would suggest to start with: "In the advance season the modeled sea ice trends diverge ..."**

Response – The text has been updated as suggested.

From: The advance season is when the CMIP5 models sea ice trends diverge most significantly from observed trends, particularly in the Ross/Amundsen and Amundsen/Bellingshausen sectors where the highest-magnitude change is also observed (Hobbs et al., 2015; Hobbs et al., 2016).

To: In the advance season, the modelled sea ice trends diverge most significantly from observed trends, particularly in the Ross/Amundsen and Amundsen/Bellingshausen sectors where the highest-magnitude change is also observed (Hobbs et al., 2015; Hobbs et al., 2016).

In contrast to the rest of the manuscript, I find the conclusion section a bit weak. I think it hides some major findings that are more clearly stated in the results section. I would also love to see that the last sentence/paragraph contains the major conclusion(s) of or the overall benefit from the present paper, rather than an outlook as it is currently done. To me, this leaves the impression the results of this paper are not important which is not true.

Response – The conclusion section has been reorganised and rewritten in parts to more clearly state the conclusions and their importance in context of other academic literature.

**Fig.1 and Fig.S1 (captions): I would prefer red dotted/blue "contours" or "isolines" instead of just "lines".**

Response – This has been updated as suggested.

**Fig.2 (caption): The authors mention dotted lines at 0.4 and -0.4. I cannot find them in the figure.**

Response – Figure 2 has been updated to show dotted lines at 0.7 ( $r^2=50\%$ ) and 0.9 ( $r^2=80\%$ ) for ease of interpretation. At the advice of Reviewer 1, the dotted lines have not been extended to negative correlations.

**Fig.5: It would be very helpful for the reader if the authors would use the same color for each model as in Fig.3. I cannot see a reason for not doing so.**

Response – The colour scheme here has been updated to match the other coloured plots.

Technical comments: p.2, l.19: "i" is missing in comparatively p.3, l.33: remove one "boundaries" p.5, l.10: significance instead of "significant" Fig.6 (caption): a dot is missing after "retreat"

Response – All these technical comments have been implemented in the text.

[revised manuscript text omitted]

the inverse sign to the observations. These comparisons only measure the extent to which the observed spatial pattern was replicated in the models, not whether the magnitude of the interactions in the models is similar to that of the observations. The correlation between simulated patterns and observed patterns during advance is plotted horizontally, while the correlations during retreat are plotted vertically for each sector. (Correlation maps for individual models can be seen in Figure \$2).

Simulated SIE and SLP correlation patterns most closely reflect observed patterns during the season of advance. The percentage of variance in the observed pattern that can be accounted for by each simulated pattern can be obtained by calculating the coefficient of determination, r2, which is the square of the pattern correlation value. During advance, 5 of the 16 models simulate a correlation pattern in the Ross/Amundsen sector that can account for at least 80% of the spatial variance in the observed pattern, while 12 of the 16 models simulate a pattern that can account for over 50% of the observed pattern. Correlation pattern with an r2 value of at least 80% of the observed pattern, and 13 of the 16 models simulations with an r2 value of over 50% of the observed pattern. For East Antarctica and King Hakon VII, the number of model simulations with patterns that can account for at least 50% of the observed pattern is 12 and 4 respectively.

However, during the retreat season, the simulated patterns are less consistent with the observed patterns. Only in the Weddell sector do more simulations produce patterns that can account for over 50% of the pariance in the observed pattern during retreat (5) than in advance (0). In the Ross/Amundsen sector, simulations during ice retreat continue to reflect the observations reasonably well, though not as strongly as in advance. In the remaining sectors, especially East Antarctica and the Amundsen/Bellingshausen, the simulations largely do not capture the observed SIE and SLP correlations during retreat.

15

20

These results have shown that the models have varying levels of success in representing the atmosphere's impact on sea ice variability. It is particularly interesting that the models reproduce these atmosphere-sea ice interactions more strongly during the period of advance than during retreat, especially given the strong representations in the Ross/Amundsen and Amundsen/Bellingshausen sectors, and that ice advance is the period during which model trends of SIE in these sectors deviate most significantly from the observed trends (Hobbs et al., 2015; Hobbs et al., 2016). Given the discrepancy between simulated and observed SIE trends, it is pertinent to consider whether the extent to which models represent observed atmospheric variability also impacts upon their representation of sea ice trends. To examine this issue, the same pattern correlation values discussed above are plotted for each model against that model's SIE trend for that sector and season, which is calculated using

30 the ensemble average of the model's historical simulation (Figure 3). The observed trend for each sector and season is plotted as a red dotted line. There does not appear to be a strong relationship between, higher pattern correlation values (indicating close agreement between the model correlation maps and that of the reanalysis) and the proximity of model SIE trends to observed SIE trends in each sector and season. This is most clearly noticeable in the Ross/Amundsen, the Amundsen/Bellingshausen and East Antarctica, particularly during advance (Figures 3a, 3c, and 3i). In these sectors, although

9

|
I |        | C • |
|-------|--------|------------|
| 110   | i noto | ~ .        |
|       |        |            |

| Deleted: explained     |
|------------------------|
|                        |
| Deleted: explains      |
| Deleted: explains      |
| Deleted: retreat       |
| Deleted: that explains |
| Deleted: explaining    |
| Deleted: explaining    |
|                        |

the representation of the reanalysis correlations is generally strong, a wide spread in trend values is also evident. These results suggest that a model with an interannual sea ice-atmosphere interaction pattern that closely represents the observed pattern will not necessarily also produce realistic sea ice trends.

**4.3 Model representation of large-scale atmospheric modes**

- 5 The leading atmospheric mode produced by the EOF analysis of ERA-Interim SLP data clearly displays the spatial pattern of the circumpolar SAM and the associated ASL, explaining 36% of the variance in SLP during advance and 40% during retreat (Figure 4a and 4b). The second and third eigenvectors illustrate the spatial pattern of the PSA (Mo and Ghil, 1987). These two PSA modes were added together to produce a single mode representing the influence of tropical forcing on the high southern latitudes, in order to compare observation-based and simulated tropical impacts on sea ice (Figure 4c and 4d). The combined
- 10 PSA EOF accounts for 27% of the variance in SLP during advance and 21% during retreat. The EOF analysis was then conducted on the individual ensemble members of each model, revealing the forced climate response of each model member. Individual ensemble member EOFs can be seen in Figure \$3. These were then correlated with the EOFs from ERA-Interim (Figure 5).
- 15 The results are explained using two metrics. The first metric, correlation values, is used to indicate the strength of the simulated representation of the spatial pattern seen in the reanalysis. A correlation value close to 1 indicates good representation of the pattern, while values near 0 indicate little resemblance between the two. A second metric was created by dividing the amount of atmospheric variance explained by the model EOF by the amount of variance explained by the observation-based pattern, creating a ratio of the percentage of variance explained. A ratio of 1:1, which would appear on the dotted curved reference
- 20 line, would indicate that the amount of variance explained by the pattern in the models is the same as the amount explained in the observation-based pattern, while a higher or lower ratio, appearing above or below the dotted reference line, would indicate whether the model is over-representing or under-representing the influence of this atmospheric pattern.
- The first EOF shows loose clustering across the ensemble members, indicating general agreement within individual models in their representation of the spatial pattern of the SAM during both ice advance and ice retreat (Figures 5a and 5b). Of the 73 individual ensemble members used in the study, 68 during advance and 45 during retreat produced a reasonable spatial pattern of the SAM as evidenced by correlation values greater than 0.7. No ensembles during either advance or retreat obtained correlation values of 0.5 or less. In terms of the percentage of atmospheric variance explained by the simulated patterns compared with that of the observation-based pattern, the patterns of 45 ensemble members during advance and 60 during
- 30 retreat account for a ratio of variance higher than the 1:1 ratio that indicates agreement with the variance explained by the reanalysis. This shows that the relative influence of SAM is overestimated in a large proportion of models, particularly during the season of ice retreat, consistent with Haumann et al. (2014). The response of sea ice to SAM is stronger during retreat than

| 1 | $\mathbf{n}$ |
|---|--------------|
|   |              |
|   | v            |

| - | Deleted: This suggests                                                                 |
|---|----------------------------------------------------------------------------------------|
| - | Deleted: the ability of the models to simulate correlations between SIE and SLP |
| Ì | Deleted: reflect                                                                       |
| Ì | Deleted: correlation patterns does                                                     |
| Ì | Deleted: mean that models                                                              |
| ľ | Deleted: SIE trends that reflect observed SIE                                          |
| - | Deleted: just over                                                                     |

| - | Deleted: just over            |
|---|-------------------------------|
| - | Deleted: historical ensembles |
| - | Deleted: the models.          |
| - | Deleted: model                |
| 1 | Deleted: S2                   |
| ή | Deleted: Correlation          |

[revised manuscript text omitted]
 entre the close representation of observed interannual entre the close representation of observed interannual entre the close representation entre the close repr
- 30 this season.

[revised manuscript text omitted]

5 supported by the Australian Government's Cooperative Research Centres Programme through the Antarctic Climate and Ecosystems Cooperative Research Center (ACE CRC), and contributes to AAS Project 4116.

Table 1: Summary of models from the Coupled Model Intercomparison Project Phase 5 (CMIP5) used in the study, showing the Institution/Modelling Centre and official model name

| Modelling Centre/Group                                          | Model Name                              |
|-----------------------------------------------------------------|-----------------------------------------|
| Commonwealth Scientific and Industrial Research Organization    | ACCESS1.0                               |
| (CSIRO) and Bureau of Meteorology (BOM), Australia              | (Bi et al., 2013)                       |
| CSIDO and DOM Australia                                         | ACCESS1.3                               |
| CSIKO aliu BOW, Ausualia                                        | (Bi et al., 2013)                       |
| Beijing Climate Center, China Meteorological Administration     | BCC-CSM1.1                              |
|                                                                 | (Xiao-Ge et al., 2013)                  |
| Canadian Centre for Climate Modelling and Analysis              | CanESM2                                 |
|                                                                 | (Arora et al., 2011)                    |
| National Contar for Atmognharia Daggarah                        | CCSM4                                   |
| National Center for Aunospheric Research                        | (Gent et al., 2011)                     |
| Community Forth System Model Contribution                       | CESM1-CAM5                              |
| Community Earth System Model Controlitors                       | (Neale, 2010)                           |
| Centre National de Recherches Météorologiques / Centre Européen | CNRM-CM5                                |
| de Recherche et Formation Avancée en Calcul Scientifique        | (Voldoire et al., 2013)                 |
| LASG, Institute of Atmospheric Physics, Chinese Academy of      | FGOALS-g2                               |
| Sciences and CESS, Tsinghua University                          | (Li et al., 2013)                       |
| NOAA Geophysical Fluid Dynamics Laboratory                      | GFDL-CM3                                |
|                                                                 | (Griffies et al., 2011)                 |
| Institut Diarra Simon Lanlaga                                   | IPSL-CM5A-LR                            |
| ilistitut i leite-Sillion Laplace                               | (Mignot and Bony, 2013)                 |
| Institut Pierre-Simon Laplace                                   | IPSL-CM5A-MR                            |
|                                                                 | (Mignot and Bony, 2013)                 |
| Atmosphere and Ocean Research Institute (The University of      | MIROC5                                  |
| Tokyo), National Institute for Environmental Studies, and Japan | (Watanabe et al. 2010)                  |
| Agency for Marine-Earth Science and Technology                  | (************************************** |
| Max-Planck-Institut für Meteorologie (Max Planck Institute for  | MPI-ESM-LR                              |
| Meteorology)                                                    | (Jungclaus et al., 2013)                |
| Max-Planck-Institut für Meteorologie (Max Planck Institute for  | MPI-ESM-MR                              |
| Meteorology)                                                    | (Jungclaus et al., 2013)                |
| Meteorological Research Institute                               | MRI-CGCM3                               |
|                                                                 | (Yukimoto et al., 2012)                 |
| Norwegian Climate Centre                                        | NorESM1-M                               |
|                                                                 | (Bentsen et al., 2012)                  |

Figure 1: Cross-correlations (significant at 95%) of observed SIE with ERA-Interim SLP from 1979-2014 during advance (a,c,e,g,i) and retreat (b,d,f,h,j). Red dotted contours indicate negative correlations, where a decrease in sea level pressure is associated with an increase in sea ice extent; blue contours indicate positive correlations, where a decrease in sea level pressure is associated with a decrease in sea ice extent. Black lines show sector boundaries.

Figure 2: Pattern correlation values comparing observations and CMIP5 piControl ensemble correlation maps of SLP and SIE in 5 the (a) Ross/Amundsen Seas; (b) Amundsen/Bellingshausen Seas; (c) Weddell Sea; (d) King Hakon VII; and (e) East Antarctica sectors. Dotted lines at 0.7 and 0.9 show the point at which the coefficient of determination, r2, is equal to 50% or 80%, respectively. The diagonal line indicates where correlations for both seasons would be in agreement.

I